# Investigating the internal structure of the Antarctic Ice Sheet: the utility of isochrones for spatiotemporal ice-sheet model calibration

Johannes Sutter[1], Hubertus Fischer[1], and Olaf Eisen[2,3]

[1]Climate and Environmental Physics, Physics Institute, and Oeschger Centre for Climate Change Research, University of Bern, Bern, Switzerland
[2]Alfred Wegener Institute Helmholtz-Centre for Polar and Marine Research, Bremerhaven, Germany
[3]Department of Geosciences, University of Bremen, Bremen, Germany.

**Correspondence:** Johannes Sutter (johannes.sutter@climate.unibe.ch)

**Abstract.** Ice sheet models are a powerful tool to project the evolution of the Greenland and Antarctic Ice Sheets, and thus their future contribution to global sea-level changes. Testing the ability of ice-sheet models to reproduce the ongoing and past evolution of the ice cover in Greenland and Antarctica is a fundamental part of every modelling effort. However, benchmarking ice-sheet model results against real-world observations is a non-trivial process, as observational data comes with spatiotemporal gaps in coverage. Here, we present a new approach to assess the accuracy of ice-sheet models which makes use of the internal layering of the Antarctic Ice Sheet. We calculate isochrone elevations from simulated antarctic geometries and velocities via passive Lagrangian tracers, highlighting that a good fit of the model to two dimensional datasets such as surface velocity and ice thickness, does not guarantee a good match against the 3D architecture of the ice sheet and thus correct evolution over time. We show that paleoclimate forcing schemes derived from ice core records and climate models commonly used to drive ice-sheet models work well to constrain the 3D structure of ice flow and age in the interior of the East Antarctic Ice Sheet and especially along ice divides, but fail towards the ice-sheet margin. The comparison to isochronal horizons attempted here reveals that simple heuristics of basal drag can lead to an overestimation of the vertical interior ice-sheet flow especially over subglacial basins. Our model-observation intercomparison approach opens a new avenue for the improvement and tuning of current ice-sheet models via a more rigid constraint on model parameterisations and climate forcing, which will benefit model-based estimates of future and past ice-sheet changes.

## 1   Introduction

A core motivation of the ice-sheet modelling community is to provide meaningful projections of future sea-level rise. Towards this goal, great strides have been made in improving ice-sheet models (ISMs) to capture current and past modes of ice-sheet dynamics (Pattyn, 2018). However, at this point projections of the future behaviour of the Greenland and Antarctic Ice Sheets under global warming are divergent depending on model physics, initialisation and the choice of climate forcing (Seroussi et al., 2019, 2020; Goelzer et al., 2020). This has important ramifications for model-based estimates of future sea-level change (e.g. Edwards et al., 2021) especially in the case of the Antarctic Ice Sheet due to its potentially highly nonlinear response to climate warming (e.g. Garbe et al., 2020). Despite major improvements, ISM simulations today are constrained by incomplete

boundary conditions such as uncertainties in the bedrock relief and geothermal heat flux, fragmental information on past and

present climate conditions (especially with regard to ocean circulation underneath ice shelves) and model heuristics such as parameterisations of grounding-line dynamics, ice flow and ice-shelf calving processes. To overcome these limitations, ISMs are usually tuned to observations of the past and present state of an ice sheet. Well established two dimensional data benchmarks used in ice-sheet modelling are e.g. the BEDMAP2 (Fretwell et al., 2013), BedMachine (Morlighem et al., 2017, 2020) and MEaSUREs (Joughin et al., 2015, updated 2018; Mouginot et al., 2017, updated 2017) datasets for Antarctica and Greenland,

which provide a continental coverage of bed topography, present-day ice-sheet geometry and surface flow. Such benchmarks are necessary to accurately reproduce the currently observed geometry and velocity. Ideally, ISMs are not only tuned to the present state of an ice sheet but also to paleo-horizons to ensure their capability to capture an ice-sheet's evolution under climate conditions differing from the Common Era (e.g. DeConto and Pollard, 2016; Sutter et al., 2019; Seroussi et al., 2019; Edwards et al., 2019). Typical target climate epochs are the Last Glacial Maximum (LGM) with relatively well constrained

grounding-line margins in Antarctica (Bentley et al., 2014) and paleo ice-sheet elevation proxies that can be utilised as a tuning target (e.g. Albrecht et al., 2020). Furthermore, sea-level reconstructions from the Last Interglacial (LIG) can give an indication of ice loss in Antarctica and Greenland (Dutton et al., 2015). However, lack of spatial datasets for past climate states as well as uncertainties and sparse coverage of paleo-proxy based ice-sheet and sea-level reconstructions complicate the validation of paleo ice-sheet simulations.

Using the above-mentioned tuning targets allows for simulation of an ice sheet in line with the present-day observed surface properties and proxy data from the past millennia. However, the notion that a good fit to spatial datasets of the Common Era and proxy targets in the past guarantees the model's ability to respond accurately to future climate changes is debatable. Due to the complexity of ice-sheet-climate interactions it is still challenging to create a proper initial ice-sheet configuration from which its future evolution can be adequately simulated (Seroussi et al., 2019, 2020). For example, tuning the ice sheet to the observed

present state (e.g. via inversion for basal drag) by matching the current ice-sheet topography and surface flow does not guarantee the accurate reproduction of e.g. internal flow, ice temperature distribution and basal friction. It also could lead to overfitting of parameters relevant to ice flow within the scope of uncertain boundary conditions such as geothermal heat flux, sub-shelf ocean temperatures, and surface mass balance. Without inversion the modelled present-day topography can differ from the observed state by several hundred metres of ice thickness on which basis it is difficult to interpret model-based sea-level projections.

Fundamentally, every ISM application is an ill-posed problem with non-unique solutions. Therefore, overfitting to a set of observables could lead to an initial ice-sheet configuration dominating the projected response to the applied climate forcing (Seroussi et al., 2019), especially over decadal to centennial timescales. Discrepancies in the initial state with respect to the actual real world ice sheet can propagate and multiply during the model simulation due to the intrinsic nonlinearities of the system. Even a near-perfect match to present-day 2D or 1D observable state variables can conceal overfitting of the model

due to weakly constrained boundary conditions, e.g. uncertainties in the climate forcing or geothermal heat flux (Burton-Johnson et al., 2020; Talalay et al., 2020). To counter the effects of overfitting, promising attempts to improve the initialisation of ice sheet models have been made which involve transient inversions of multiple surface elevation observations over time (Goldberg et al., 2015) albeit only on the regional scale and limited to the extent of the satellite record. To reduce the effect of

these limitations, we propose an additional data benchmark for ISMs which provides spatiotemporal constraints incorporating the paleo-evolution of an ice sheet and complements the hitherto used 2D and 1D tuning targets.

Paleo-surfaces within ice sheets are created by simultaneous deposition of impurities at the ice-sheet's surface. These are widely observable by radio-echo sounding. The internal horizons (isochrones) of ice sheets provide a spatiotemporal calibration target for ISMs, as they are formed by the three dimensional ice flow patterns affected by bedrock undulations, geothermal heat flux and the paleo-climate and therefore integrate all aspects of an ice-sheet's evolution in one observable. Isochrones are mapped by radar observations as individual radar reflections within the ice and are subsequently dated for example by using tie points to ice core chronologies (crossover points) or distinct reflectors such as dated volcanic eruptions. An ice-sheet wide coverage of such traced and dated englacial layers, as envisioned by the SCAR action group AntArchitecture (Bingham, 2020) would be an invaluable tuning benchmark, which both constrains the climate forcing as well as the simulated ice flow more tightly than the usually employed 2D or 1D data targets. So far, isochrones have been used in or with ice-flow models (1D, 2D or 3D) to reconstruct paleo accumulation patterns and their changes, e.g. in Waddington et al. (2007); Leysinger Vieli et al. (2011); Karlsson et al. (2014); Cavitte et al. (2018), or to constrain high resolution alpine glacier models (Konrad et al., 2013; Jouvet et al., 2020). Specialised models have been developed to compute isochrone geometry (Hindmarsh et al., 2009) or tailor-made to accurately simulate the internal layering of ice sheets (Born, 2017). Lagrangian and semi-lagrangian tracer advection has been employed to shed light on the isotopic composition and stratigraphies of the Greenland and Antarctic Ice Sheets (Clarke et al., 2003, 2005; Lhomme et al., 2005a, b; Huybrechts et al., 2007; Goelles et al., 2014; Born, 2017), to identify potential locations of old ice in Antarctica (Passalacqua et al., 2018) and recently to assess past ice-sheet instabilities (Sutter et al., 2020). Despite these achievements, englacial isochrones have not been getting the required attention in the context of tuning targets for continental ISMs (Born, 2017).

In this work, we discuss the ability and limitations of a 3D continental ISM to capture englacial layers and therefore to exploit existing isochrone elevation datasets to constrain ISM results. We compare model results to dated isochrones spanning a large part of the East Antarctic Ice Sheet and discuss how such a comparison can be utilised to improve ISM simulations. Thus, this work does not aim at reconstructing the internal architecture of the Antarctic Ice Sheet in as detailed a way as possible (as e.g. done locally in Parrenin et al., 2017), but rather to exploit isochrones as a constraint on model behaviour to improve the modelling of paleo-ice-sheet evolution. In section 2 we introduce the post-processing routine (Lagrangian tracer advection) and the input data, give an overview of the observed traced and dated internal layers from East Antarctic radar transects discussed in this study and summarise the ice-sheet model setting. In section 3 we focus on the ice-core deep drilling site Dome C and provide a detailed analysis of the simulated internal layering in comparison to a set of observed layers from four individual radar transects, discussing paleo-accumulation patterns the impact of ice dynamics as well as uncertainties in bedrock elevation and geothermal heat flux. Finally, we expand this view to the whole East Antarctic Ice Sheet in section 4, where we highlight regions where our model-isochrone comparison shows systematic deficits in the modelled ice flow and suggest processes and parameterisations in the model that may be responsible for these deficits.

## 2 Approach to modelling Antarctic isochrones

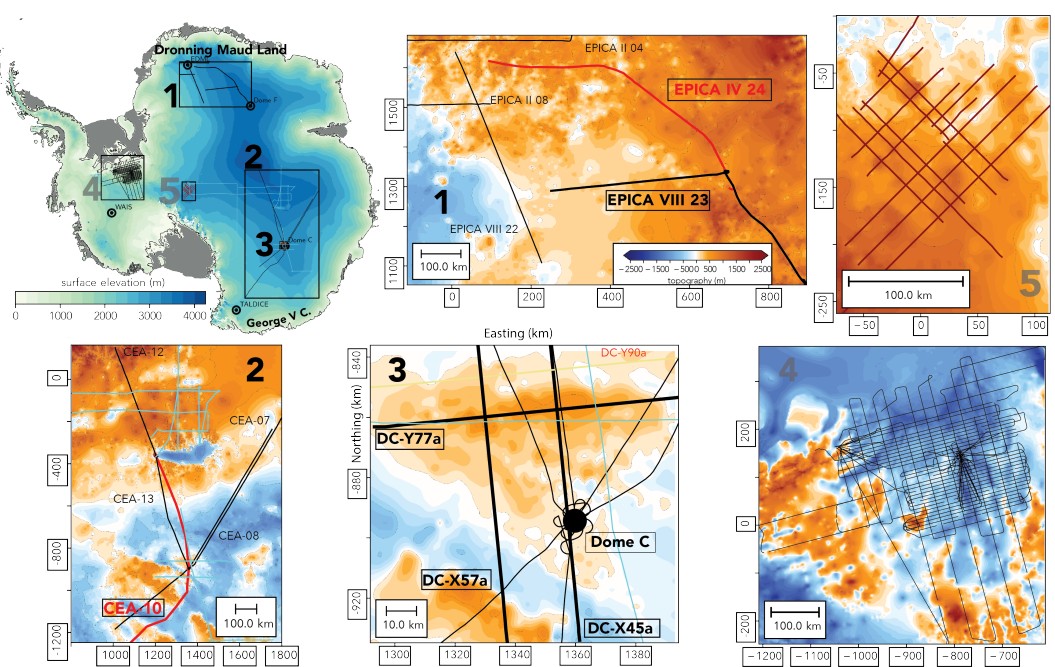

**Figure 1.** Antarctic surface (top left) and bedrock elevation from BedMachine Antarctica (Morlighem et al., 2020) overlain by radar transects with individual dated isochrones analysed in this study. The enlarged sections highlight regional clusters and their respective topographic settings. Clusters 1-3 are discussed in this study. The isochrone data are from: cluster 1) Winter et al. (2019), cluster 2) and 3) Leysinger Vieli et al. (2011); Cavitte et al. (2016, 2020); Winter et al. (2019), cluster 4) Beem et al. (2020), cluster 5) Ashmore et al. (2020). Northing/Easting is attained via cartographic transformation based on the WGS84 reference coordinate system.

We use ISM results from Sutter et al. (2019) as well as a new set of paleo-simulations and a present-day equilibrium ensemble obtained from simulations with the ISM PISM (the PISM authors, 2015; Bueler and Brown, 2009; Winkelmann et al., 2011b) to compute Antarctic isochrone elevations. We tune the ISM to match observational data from present day (PD), the Last Glacial Maximum (LGM) and the Last Interglacial (LIG). Tuning targets include the surface elevation at East Antarctic ice core locales as well as the grounding-line extent at the LGM (Bentley et al., 2014), LIG sea-level reconstructions (Dutton et al., 2015) and PD sea-level equivalent ice volume and grounding-line position (Fretwell et al., 2013). We choose selections of traced and dated englacial layers that have crossovers with the EPICA Dome C (EDC), EPICA Dronning Maud Land (EDML), Dome Fuji and Vostok ice-cores. Unfortunately, the number of published and available traced and dated Antarctic isochrones is limited at the current stage, a situation which initiatives like AntArchitecture aspire to improve in the coming years. For this intercomparison we make use of three compilations of dated isochrones (Leysinger Vieli et al., 2011; Cavitte et al., 2016, 2020; Winter et al., 2019) which provide about 10,000 km of analysed radar transects with dated isochrones covering the past 38 ka to 170 ka BP (see Table 1). We focus on this time period as dated isochrones from this range are available for all regions considered

**Table 1.** Overview of analysed isochrones. Transects presented in this work are DC - X45a, X57a,Y77a,Y90a, CEA-10, DML VIII-23/IV-24.

| Name | Region | Line length (km) | Age (ka) |
|---|---|---|---|
| [C]DC - **X45a,X57a,Y77a,Y90a** | Dome C | 100 | 38,48,73,96,130,160 |
| [W]CEA-7,8,**10**,12,13 | Wilkes Basin - Dome C - Vostok - Dome A | 400 - 1800 | 38,90,161 |
| [W]DML- II-04,II-08,**IV-24**,VIII-22,**VIII-23** | Dome Fuji - EDML | 200 - 800 | 38,74 |
| [LV]DC - 1,2,3 | Dome C | 50 - 600 | 36.5,73,130,170 |
| [LV]V - 1,2,2W,3 | Vostok - Dome A - South Pole | 50 - 2000 | 36.5,73,130,170 |
| B | Titan Dome | $\approx 100$ km | 4.7, 10.7,16.8,29,37.6,51.4,72.5 |
| A | Weddell Sea sector West Antarctica | $\approx 100 - 600$ km | 3.1,5.6,6.4 |

Isochrones are from Cavitte et al. (2016, 2020) ([C]DC-X45a,X57a,Y77a,Y90a), Winter et al. (2019) ([W] CEA-7,8,10,12,13,DML- II-04,II-08,IV-24,VIII-22,VIII-23), Leysinger Vieli et al. (2011) ([LV] DC - 1,2,3, [LV] V - 1,2,2W,3), Beem et al. (2020) and Ashmore et al. (2020). The superscripts ([C,W,LV,B,A]) denote the respective publication. The line lengths (x-y) specified in the table vary depending on the transect except for the data from Cavitte et al. (2016, 2020) where all transects are 100 km in length. The specified ages are those used in this manuscript but do not represent the complete set of internal layers available in the respective references. The transects shown in the lower half of the table are not discussed in this work.

here. Both younger and older dated isochrones are available for example for Dome C, Titan Dome (Beem et al., 2020) and for West Antarctica (Ashmore et al., 2020) (see Table 1). With this wealth of information we are able to not only analyse the performance of our model simulations close to ice core locations but also along and across ice divides spanning the whole distance from Dronning Maud Land to the George V Coast (see Figure 1). The area connected by the radar transects covers a wide range in the spectrum of bedrock relief, geothermal heat flux as well as climate regimes and ice-sheet elevations. This allows us to analyse the impact of uncertainties in the paleo mass balance, basal melt, ice dynamics and bedrock on modelled isochrone elevations. Table 1 gives an overview of the radar transects analysed in this work. The Dome C region is included in Leysinger Vieli et al. (2011); Winter et al. (2019); Cavitte et al. (2020) and provides an excellent testbed for the validity of the climate forcing scheme employed by Sutter et al. (2019). The EDC ice core provides tight constraints on the age of the isochrones via the AICC2012 chronology (Veres et al., 2013; Bazin et al., 2013) as well as the accumulation history during the last glacial-interglacial cycle (Cavitte et al., 2018). It is important to note that, as the ISM's climate forcing was modulated via the EDC deuterium record, we expect the best fit of our computed isochrones to be in the region encompassing this ice core. The areas around EDC, Dome Fuji and EDML are all well sampled by reliable radar observations, therefore providing very accurate estimates of bedrock elevation (Karlsson et al., 2018). This makes it possible to investigate the impact of bedrock uncertainties on the mismatch between observed and computed isochrone elevations. Finally, the transects in the vicinity to faster flowing marine drainage regions (e.g. Wilkes Subglacial Basin) allow for an assessment of model parameterisations of ice flow on the computed isochrone elevation.

### 2.1 Observation of isochrones

Radar surveys of ice sheets observe numerous internal layers, i.e., changes in the dielectric properties of the ice, which lead to a partial reflection of the downward propagating radar wave back to the surface. Most of these layers are caused by changes in

the conductivity content of the ice by deposition of acids from volcanic eruptions at the surface of the ice sheet. After volcanic eruptions these depositions usually occur over a few years and are rather homogeneous over spatial scales of 10s to 100s of km. The absence of melting on the Antarctic plateau leads to continuous burial and, thus, submergence of these conductivity layers, which are typically spread over less than one meter (Eisen et al., 2006). Lateral variations in accumulation rates cause differences in submergence velocities and, thus, to deformation of these initially surface-parallel layers. With increasing depth,
the influence of ice dynamics on the (vertical) submergence velocity increases and deforms the layers further.

Radar systems used to map internal layers and the thickness of ice sheets operate in the range of 10s to 100s of MHz, corresponding to a wavelengths on the order of metres. The width of the observed reflections, caused by the interaction of the electromagnetic radar waves with the conductivity peaks, is determined by the bandwidth or pulse length of the radar systems. They are typically in the range of 0.5 to several tens of metres (Winter et al., 2017).

Occurences of basal melt, e.g. due to high local geothermal heat flux, can cause deformation of the internal layers closer to the bed. With increasing depth the internal layers follow the bed topography in an often smoothed fashion. Thus, the lateral variation of the depth of these internal reflection horizons is usually sufficiently small to be continuously detected with the spatial sampling distance of airborne radar systems. They operate with a horizontal sampling distance in the range of metres to tens of metres (Winter et al., 2017) and can be traced over hundreds to thousands of kilometres. As each of these layers
originates from the same physical change in conductivity, which the ice inherited during deposition as snow at the surface, they are considered isochrones and have the same age along the reflection horizon.

Dating can most reliably be achieved by tying internal layers to ice core sites, where age–depth estimates are available from ice-core proxy analysis. For the comparison of observed and modelled isochrone elevations, the uncertainty of the observed isochrones is an important parameter. Apart from the radar-inherent uncertainties associated with identifying and continuously
tracing the same radar signal for an horizon and the conversion of two-way travel time to depth, the uncertainty in the ice-core ages and the actual matching procedure between horizons and ice core are the largest sources of error.

We use various internal layers in this study. Details on the radar resolution, dating methods and resulting uncertainty can be found in the respective original publications (Cavitte et al., 2016; Winter et al., 2019; Cavitte et al., 2020; Leysinger Vieli et al., 2011). Based on the analysis performed on five different radar systems by Winter et al. (2017) near EDC, we consider
a maximum age uncertainty of around 1 ka for each isochrone above 2000 m depth (roughly 2/3 of the ice thickness) (Winter et al., 2019). This range covers most of the horizons considered in this study. Below a depth of 2000 m, the age uncertainty increases non-linearly with depth towards the bed (Winter et al., 2019). At places where the deepest ice is younger than the ice around Dome C, the age gradient with depth will then be less steep towards the bed than the one determined at Dome C. Thus, the age uncertainties of the horizons below 2000 m depth will be lower than at Dome C. We consider this uncertainty always
to be smaller than the proposed age derived from the ISM data.

## 2.2 Large-scale ice-sheet modelling

In order to compute Antarctic isochrones, time-resolved 3D velocity data as well as the transient ice-sheet geometry (ice thickness and bedrock topography) are necessary. We therefore ran a paleoclimate model ensemble covering the last 220 ka.

All 220 ka simulations were initialised from the 220 ka output of a 2 Ma long simulation in Sutter et al. (2019). We also make use of four simulations from Sutter et al. (2019) which are based on four different geothermal heat flux fields (Shapiro and Ritzwoller, 2004; Purucker, 2013; An et al., 2015; Martos et al., 2017). In addition to the 220 ka paleo-ensemble we carried out a present-day equilibrium ensemble to assess the impact of the missing paleo-spinup as well as different model parameterisations on the computed isochrone elevations.

Isochrone elevations (see 2.3) are computed on the basis of : 1) full paleo-ISM runs (called pal) in which model integration starts from the 220 ka time slice of a 2 Ma simulation (Sutter et al., 2019), 2) the present-day snapshot of the 220 ka simulation (pd-pal), and 3) a present-day equilibrium ensemble (pd) with an integration time of 2000 years following a thermal spinup using a fixed ice sheet geometry for 200 ka. The 2 Ma simulations in Sutter et al. (2019) are initialised at 2 Ma BP from a present-day ice sheet geometry. Isochrone evolution starts at the isochrone's respective age (see Table 1) in the past and follows the computed transient ice flow. In the present-day snapshot (pd-pal) and present-day equilibrium ensemble (pd) isochrones evolve on the basis of the simulated present-day flow (constant velocity field).

The climate forcing for the present-day ensemble was derived from the regional climate model RACMO (van Wessem et al., 2014) and the World Ocean Atlas 2018 (Locarnini et al., 2019). The parameter range chosen for the PD ensemble is associated with an equilibrium sea-level equivalent ice volume within $\pm 2$m of present-day observations (Fretwell et al., 2013). Both the paleo simulations and the present-day equilibrium simulations were tuned to match the observed present-day surface elevation (with a focus on the deep ice core sites), ice volume and grounding-line position. An overview of the experimental setup is provided in Figure 2. The model forcing used in the paleo-simulations is described in detail in Sutter et al. (2019) and provides spatiotemporal information on ocean temperatures, surface temperature and precipitation derived from climate snapshots during the LIG (Pfeiffer and Lohmann, 2016) and the LGM as anomaly forcing with respect to the modelled pre-industrial climate state. In between the climate snapshots, surface temperature and ocean temperatures are interpolated on the basis of the EDC deuterium data (Jouzel et al., 2007) using a temperature-precipitation relationship of 3 %$\mathrm{K}^{-1}$ (see section 2.3 and Figure 2 in Sutter et al., 2019) and 5, 6 and 8 %$\mathrm{K}^{-1}$ for the 220 ka simulations. The ISM is run on a 16 km grid with 81 vertical levels. Bedrock elevation changes due to transient load changes are computed via the Lingle-Clark model based on Lingle and Clark (1985); Bueler et al. (2007). We employ a combination of the shallow ice (SIA) and shallow shelf (SSA) approximation (SSA+SIA Hybrid Winkelmann et al., 2011a) with a sub-grid grounding-line parameterisation (Gladstone et al., 2010; Feldmann et al., 2014) to allow for reversible grounding-line migration despite using a relatively coarse resolution (Feldmann et al., 2014). Table 2 provides an overview of the friction and sliding parameters chosen for the 2 Ma, 220 ka and present-day model ensemble. In PISM the till friction angle $\phi$ controls the yield stress at the ice-bedrock interface which can be set to be a function of the bedrock elevation (increasing with elevation). The yield stress is determined by

$$\tau_c = \tan \phi N_{till} \tag{1}$$

**Table 2.** Overview of parameters relevant for sliding in Sutter et al. (2019) (2Ma), for the 220ka ensemble (220k) and for the present-day ensemble (PD). $\phi_1$ is the minimum (for depths below -700 m) and $\phi_2$ the maximum (for depths above 0 m) till friction angle.

| Parameter | $\phi_1$ | $\phi_2$ | q | $sia_e$ |
|-----------|----------|----------|-----|---------|
| 2Ma | 5 | 30 | 0.6 | 1.0 |
| 220k | 5 | 30 | 0.6 | 1.0 |
| PD | 2, 3, 4, 5 | 30 | 0.6, 0.75 | 1.0 |

where $\phi$ is the bedrock elevation dependent till friction angle and $N_{till}$ the effective pressure. As in Sutter et al. (2019) we choose a pseudo-plastic power law with the parameter $q$ controlling the amount of sliding via the relationship

$$\tau_b = -\tau_c \frac{\boldsymbol{u}}{u_{threshold}^q |\boldsymbol{u}|^{1-q}} \tag{2}$$

where $\tau_b$ is the shear stress, $\boldsymbol{u}$ ice velocity and $u_{threshold}$ the threshold velocity at which $\tau_b$ has the exact magnitude as $\tau_c$ (condition for sliding). The so called SIA enhancement factor in the 220ka and present-day ensemble is 1.0 as in Sutter et al. (2019). For further details regarding the ISM setup see section 2.1 in Sutter et al. (2019).

Geothermal heat flux is taken from Shapiro and Ritzwoller (2004) or alternatively from Purucker (2013); An et al. (2015); Martos et al. (2017). Surface and bedrock elevation data are from Bedmap2 (Fretwell et al., 2013) except in the present-day simulation where we use the new BedMachine-Antarctica dataset from Morlighem et al. (2020).

### 2.3 Lagrangian tracer advection

We use the open source python software OceanParcels v0.9 (Lange and van Sebille, 2017) which was originally designed to analyse ocean data but can be easily adopted for transient or steady-state ISM data. At its core, OceanParcels provides a Lagrangian particle tracking scheme which is computationally efficient and adaptable to a variety of input grids (Delandmeter and van Sebille, 2019). Individual particles tracks are computed following the advection equation (equation 1 in  Lange and van Sebille, 2017):

$$\mathbf{X}(t+\Delta t) = \mathbf{X}(t) + \int_{t}^{t+\Delta t} \mathbf{v}(\mathbf{x},\tau)d\tau + \Delta\mathbf{X}_b(t) \tag{3}$$

where $\mathbf{X}$ is position of the particle in space, $\mathbf{v}$ the three-dimensional velocity at the particles' position and $\Delta\mathbf{X}_b(\tau)$ is the change in position due to "behaviour", which does not play a role in our application as we assume that every ice parcel is passively advected with the velocity field without any additional advection terms. For further details regarding OceanParcels please refer to Lange and van Sebille (2017). For OceanParcels to work we need to provide three dimensional velocity data covering the time frame we are interested in (i.e., the last 160 ka as this is the age of the oldest isochrone we take into account)

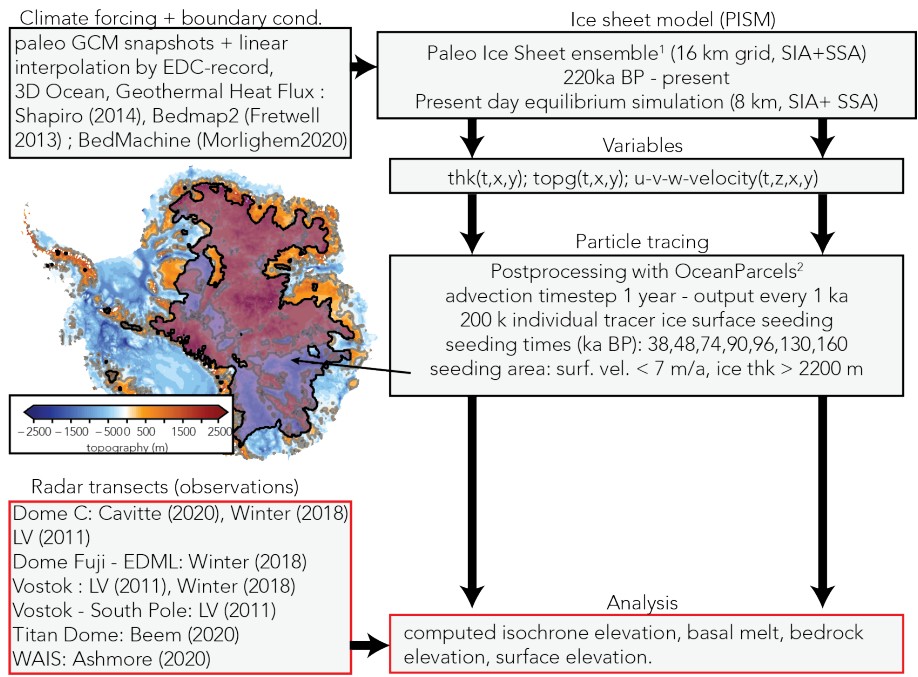

**Figure 2.** Experimental setup. The left column depicts the input data to the ISM (climate forcing, geothermal heat flux and initial geometry) and to the data intercomparison (observed radar transects). The contour plot illustrates the area where tracers are seeded (see supplementary video). The right column contains the ISM specifications, required variables (thickness, bedrock topography and 3D velocity field) for the tracing scheme and the settings used in OceanParcels (Lange and van Sebille, 2017).

as well as the ice-sheet's geometry (thickness and bedrock elevation) at sufficiently high temporal resolution. Naturally, the finer the temporal resolution of the ice-sheet time slice data, the more accurate the particle tracks will be. However, to prevent input data for OceanParcels from getting too large for standard python libraries (numpy, scipy, netCDF4) and memory we opted for a temporal resolution of 1 ka and 10 ka which proved to be accurate enough (small deviations compared to the

model uncertainties with respect to observations) for the mostly relatively slow ice flow in East Antarctica. Velocity snapshots between 1 ka and 10 ka largely produce the same isochrone elevation (see Figure 3). We set the advection time step ($\Delta t$) to one year and for each experiment we deploy 200,000 individual tracers which we seed at the ice surface at the time of the age of the isochrone we want to compute (e.g. at 38 ka BP for the 38 ka isochrone). Misfits of the model results in terms of elevation and velocity field relative to the true (unknown) ice-sheet state at that point in time in the past and throughout the

paleo simulation will lead to deviations of the modelled isochrone as observed in the ice sheet today. This information can then be used to identify such misfits and improve the model representation of the ice sheet. Here, it is important to note that observed isochrone elevations are usually defined as relative to the ice surface whereas we compute the isochrone elevation above the ice bed. Any deviations in the modelled ice bed with respect to observations will therefore imprint on the modelled isochrone elevation. Therefore, any comparison between modelled and observed isochrone elevations will be most meaningful

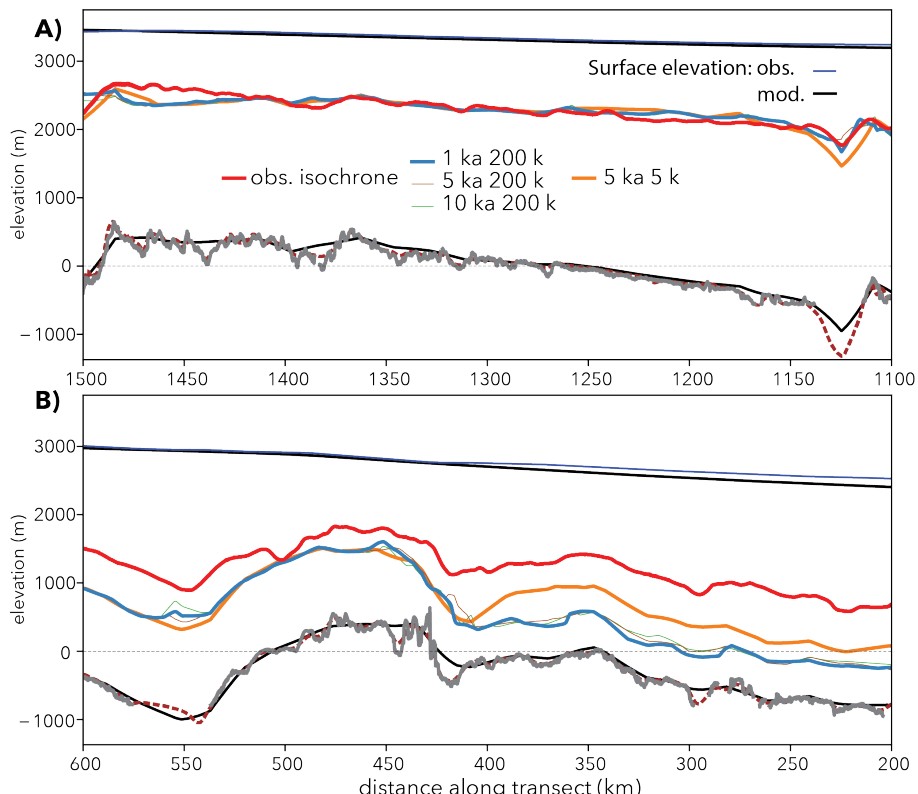

**Figure 3.** Observed and simulated isochrones along transect CEA-10 (see Figure 10 for the location of transect CEA-10). Observed isochrone is shown in red, simulated isochrones are computed based on snapshot velocity fields every 1, 5 and 10 ka. Tracer seeding was carried out with 200.000 and 5.000 tracers. Bedrock and surface elevation in the model runs is plotted in black, observed surface elevation in dark blue and bedrock elevation derived from the radar transect in grey. The gridded 1km BedMachine bedrock elevation (Morlighem et al., 2020) is plotted for comparison as well (smooth dashed red line).

along transects with small deviations between the observed and modelled bedrock elevation. We find that for less than $10^5$ tracers, gaps in coverage due to the dispersive nature of ice flow are too large to resolve the internal layers away from ice divides. Figure 3 illustrates the changes in the computed isochrone elevation due to different numbers of deployed tracers. In regions where homogenous flow dominates (see transect depicted in Figure 3 A), the number of tracers does not affect the simulated isochrone elevation much. However, if spatial gradients in ice-flow increase, simulated isochrone elevations diverge

with low tracer densities. This is shown along kms 400-200 in Figure 3 B where elevated surface velocities closer to the coast lead to more pronounced deviations of simulated isochrone elevations for a coarse seeding strategy.

The seeding mask (i.e., the region where tracers are initiated) consists of the area with an ice thickness of more than 2200 m and surface flow slower than 7 m/a (see Figure 2 and the supplementary video). The seeding mask covers an area which encloses every individual radar transect analysed here. We also tested other seeding strategies employing the IMBIE drainage

basins (Zwally et al., 2012), ice divides or around ice core locations. The choice of seeding mask is solely motivated by the tracer coverage. As long as the transects of interest are covered by the seeding mask, the latter only affects the density of tracers (a larger seeding masks leads to sparser tracer coverage if the number of tracers is not adjusted accordingly). We found that a seeding mask defined by the ice thickness and surface flow considerations mentioned above provides good coverage for the radar transects discussed in this study. The tracer experiments were all carried out on a laptop computer (Powerbook (2015),

2.7 GHz, 8 GB memory, OS-X: 10.13.6), with computation times between a few hours (38 ka isochrone) and 10-15 hours (160 ka isochrone), i.e., $\approx$ 1h/10ka elapsed model time. On a more recent machine (Powerbook (2020), 4x2.3 GHz, 32 Gb memory, OS-X: 11.3) this can be reduced to $\approx$ 10min./10ka.

OceanParcels provides netCDF-files consisting of the individual tracer positions in space and time. Accordingly, to extract isochrones we selected the last tracer positions and gridded them to a regular 1 km x 1 km grid (bi-linear interpolation). This

provided an elevation map of the respective tracer swarm for all regions covered by the seeding mask (see Figure 4 A and supplementary video). From this elevation map we then extracted the computed tracer-elevation. From the ice-sheet model output we retrieved the bedrock- and surface-elevation, the melting at the base of the ice and the corresponding geothermal heat flux (the latter being derived from the input data) along the individual radar transects. Our main goal is the identification of systematic mismatches between predicted and observed isochrone geometry. As a metric for the difference between observed

and modelled isochrones we use their respective elevations in the ice sheet above the ice-bed interface, normalised by the local ice thickness. This yields the root-mean square difference (RMSD) in %.

## 3   Modelled Antarctic isochrone elevations

Our main goal is the identification of systematic mismatches between predicted and observed isochrone geometry. To achieve this, we focus on three major sources of model-observation mismatches: (i) climate forcing, (ii) model parameterisation, (iii)

bedrock and geothermal heat flux. As computed isochrone elevations will be affected by a superposition of uncertainties from all three sources we separate our analysis into regions where we expect only one factor to dominate. To isolate the effect of (i) climate forcing we turn to areas characterised by limited ice flow (e.g. ice domes or close to ice divides) and by a good representation of the bedrock elevation in the model (difference between input dataset and local high resolution radar data). We will primarily focus on Dome C, as ice sheet model parameters relevant for ice flow and basal friction were tuned to match the

regional ice-sheet configuration. To gauge the impact of ice flow parameterisation (ii) we assess the fit of modelled isochrones in regions of elevated surface velocities and in subglacial basins (bedrock below sea level) as well as the effect of different parameter choices (see Table 2) under the same boundary conditions (climate, geothermal heat flux, bedrock topography). To test (iii) we analyse the impact on bedrock elevation uncertainty around Dome C and use different geothermal heat flux fields to investigate their influence on the modelled internal layer architecture. We would like to note that while we simulate

isochronal layers throughout East and West Antarctica our model-observation-intercomparison is mostly limited to ice domes and along ice divides in East Antarctica, as these regions are covered by dated radar-based isochrone observations. Future

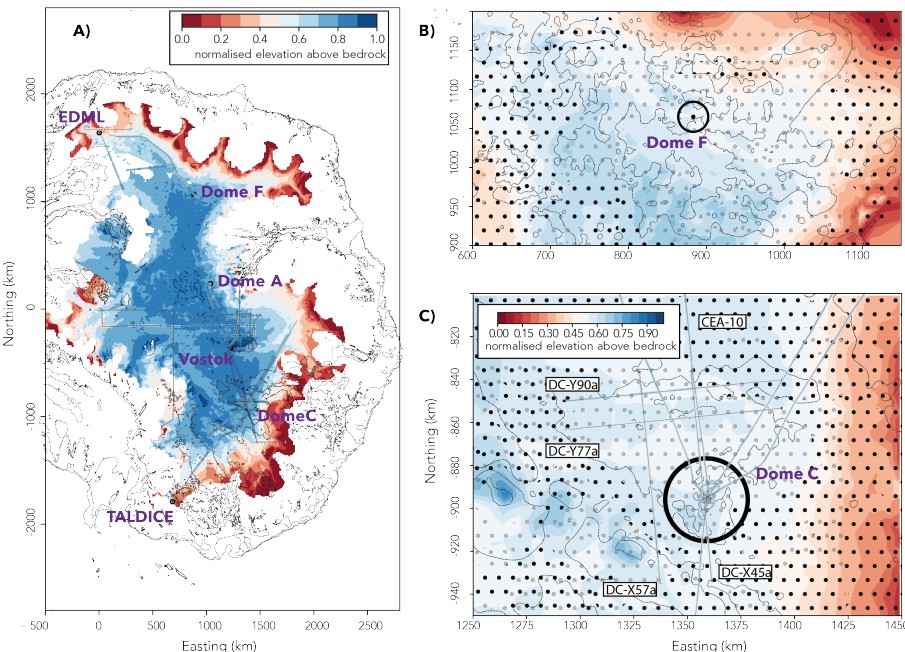

**Figure 4.** Computed isochrone elevation in East Antarctica. Panel A) illustrates the elevation above bedrock of the 38 ka isochrone normalised by the local ice thickness. The filled isochrone contours are overlain by the normalised elevation of the observed isochrones (thin coloured lines). Black shapes in A depict the zero meter sea-level bedrock contour. Panel B) shows a magnification of the region around Dome Fuji with dotted areas (filled black circles) where basal melt is simulated. Semi-transparent dots show results with geothermal heat flux forcing from Martos et al. (2017) and opaque dots from Shapiro and Ritzwoller (2004), respectively. Panel C) shows the area around Dome C overlain by the normalised elevation of the observed isochrones (thin coloured lines). The mismatch between the simulated isochronal layer and the observed isochrone elevation along the transects is generally small in the Dome C region with the exception of the upper right section in panel C) where the simulated isochrone is off by up to 40% of the local ice thickness.

radar explorations in Antarctica will hopefully complement the available data by observations away from ice divides and along drainage sectors as these are the regions which point to critical misrepresentations in ice-sheet model simulations.

### 3.1 Dome C - evaluating the paleoclimate forcing (i)

We now turn to the first (i) of the aforementioned three main factors determining simulated isochrone elevations, the climate forcing. Due to the near cessation of surface flow at ice domes the local internal stratigraphy is by and large dominated by the regional surface mass balance history and potentially affected by basal melting due to a large ice thickness, low surface accumulation and/or elevated geothermal heat flux. Consequently, in the particular geographical setting of an ice divide, we do not expect that parameterisations of ice dynamics will have a large effect on the internal layer architecture. Therefore, we
assume that model-mismatches with respect to observed isochrones are mostly due to the applied surface mass balance forcing and geothermal heat flux. To evaluate the validity of the forcing approach in Sutter et al. (2019) and in this work we turn to

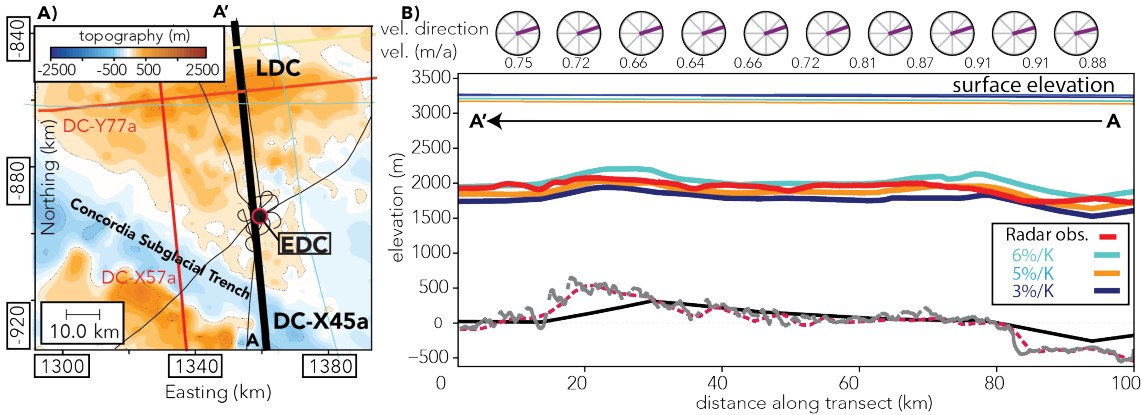

**Figure 5.** Transect DC-DC-X45awith the 96 ka isochrone. Panel A) illustrates the bedrock elevation and radar transects analysed in this study crossing Dome C. In B) the computed isochrone elevations for simulations with 3%, 5% and 6% precipitation change per Kelvin surface temperature change are plotted against the observed isochrone (red line). The modelled surface elevation is plotted in the corresponding color of the modelled isochrones and the observed surface elevation (Fretwell et al., 2013) is depicted in dark blue. Bedrock elevation in the model is plotted in black and bedrock elevation derived from the radar transect in grey. The gridded 1km BedMachine bedrock elevation (Morlighem et al., 2020) is plotted for comparison as well (smooth dashed red line). The circles on the top of B) denote the relative flow direction and speed with respect to the radar transect.

several radar transects in the vicinity of the EDC ice core ($75°1'$ S, $123°4'$ E, see Figure 5) which satisfy the condition of low ice surface velocities. It is important to note that the model parameters chosen in Sutter et al. (2019) were tuned using a climate forcing with a temperature-precipitation relationship of 3%. The paleo-simulations created for this paper employ
identical model parameters and the same forcing (see section 2.2) but with alternative temperature-precipitation relationships of 5, 6 and 8 %K$^{-1}$. The radar transects are discussed in Cavitte et al. (2016, 2020) and cut across an area loosely bound by the Concordia Trench and Little Dome C (LDC).

    In a first step we investigate which surface mass balance history during the last glacial-interglacial cycle matches best with the 96 ka isochrone along transect DC-X45a. This transect spans from the northern end of the Concordia Subglacial Trench
southward via EDC to LDC (thick black line in Figure 5 A). Isochrones are computed from the output of experiment B1-P1 in Sutter et al. (2019) using a scaling constant between temperature and accumulation anomalies (i.e. the percent change of accumulation for every degree of surface air temperature change) of 3%K$^{-1}$. Figure 5 B depicts the observed 96 ka isochrone in comparison to the simulated isochrones. The simulated isochrone elevation fits the observed one reasonably well to within $\approx 2 - 5\%$ RMSD. According to our simulations a precipitation scaling between 5 and 6 %K$^{-1}$ (RMSD of $\approx 3.8\%$ and $\approx 3.3\%$
for precipitation scaling of 5 and 6 %K$^{-1}$, respectively) reproduces the 96 ka isochrone best, which is in accordance with an ice-core-based relationship of $5.9\pm2.2\%$K$^{-1}$ for EDC (Frieler et al., 2015). This gives us confidence that the paleo mass balance forcing approach in Sutter et al. (2019) is valid at least for the region around Dome C and likely for the larger parts of the interior East Antarctic Ice Sheet (see section 4). It is important to note that the simulations are tuned to the geothermal

heat flux data from Shapiro and Ritzwoller (2004). We discuss the impact of the choice of the geothermal heat flux forcing in subsection 3.3.

Above, we identified which paleo temperature-precipitation scaling leads to the best fit of modelled and observed isochrone elevations. Making use of this paleo-accumulation estimate we can plot the transient accumulation history at EDC during the last glacial interglacial cycle (Figure 6 A) and compare the regional temporal mean precipitation forcing during 0-9 ka BP and 9-39 ka BP with reconstructed paleo-accumulation patterns at Dome C by Cavitte et al. (2018) (Figure 6 B). Figure 6 A depicts the EDC accumulation history depending on the temperature-precipitation scaling-factor. The Holocene surface mass balance fluctuates close to the present-day reference forcing derived from the regional climate model RACMO (van Wessem et al., 2014) which is used here as the present-day reference climate state (ca. 3 cm/a ice equivalent in the Dome C region) and drops to ca. 1.7 cm/a during the LGM. The observed modern surface mass balance at EDC is $\approx 2.5$ cm/a (Stenni et al., 2016), i.e., our present-day surface mass balance forcing is ca. $20\%$ too high in this region. When we compare our simplified forcing with reconstructed paleo-accumulation patterns at Dome C by Cavitte et al. (2018) our estimates of paleo-accumulation differs by about 10 to $30\%$. This is mostly due to the aforementioned overestimation in the 1979-2011 present-day reference surface mass balance at EDC. It is important to note that our approach does not strive to achieve a detailed reconstruction of paleo-accumulation as done in Cavitte et al. (2018). They are fitting englacial layer horizons via a 1-D pseudo-steady ice flow model (Parrenin et al., 2017) leading to more accurate estimates of the local surface mass balance along the ice divide near EPICA Dome C, while we are merely trying to establish the validity of our large-scale paleoclimate forcing approach. Nevertheless, Figure 6 B illustrates that our paleo-accumulation forcing provides a surface mass balance pattern along ice divides which matches more detailed reconstructions as carried out by Cavitte et al. (2018) reasonably well.

## 3.2 Dome C - impact of paleo spin-up and model parameterisation on simulated isochrone elevation (ii)

The modelled isochrone elevations discussed above were computed on the basis of transient snapshots of local velocity and topography fields (bedrock and surface elevation) and show a good match to observed isochrone elevations. In the following we analyse the modelled 96 ka isochrone-elevation along transect DC-X57a for three different tracer experiments in which the velocity and ice geometry is taken from: 1. the full transient ISM-data from the 220 ka paleo-simulation with temperature-precipitation scaling of $5\% K^{-1}$ (case pal), 2. only the present-day output of the latter (case pd-pal) and 3. the velocity and topography data from an ensemble of present-day climate equilibrium simulations (case pd) where the ice sheet is forced only by present-day surface climate and ocean temperatures. The comparison between pal and pd-pal allows us to assess the influence of the transient past surface mass balance forcing and paleo-ice-flow reorganisation in pal while the comparison between pd-pal and pd mainly illuminates the effect of the parameterisation of ice flow (ii) and the missing imprint of the paleo-spinup. In pal, tracers are advected based on the respective temporally evolving ice-sheet configuration; in pd-pal advection is based on the last time slice output (present-day) of pal (using constant velocity and topography); in pd tracers are advected via the present-day equilibrium velocity resulting from present-day climate forcing.

This will therefore elucidate the impact of the paleo-spinup and of the model parameterisation (ii) on computed isochrone elevations (Figure 7). Figure 7 shows that the difference in isochronal elevation between pal and pd-pal is already substantial.

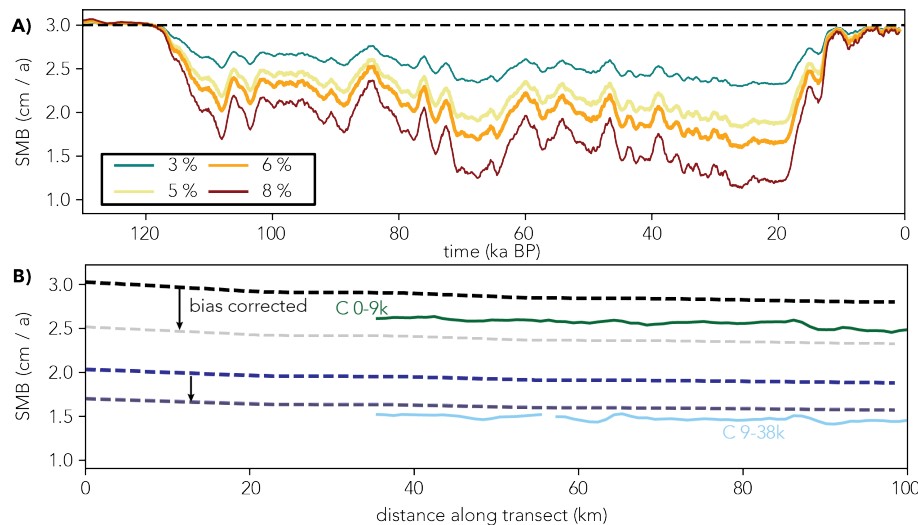

**Figure 6.** A) Prescribed precipitation (cm/a) over time at the EPICA Dome C ice core drill site for different temperature-precipitation scaling $(3, 5, 6, 8\% K^{-1})$. The temperature-precipitation scaling which produces the most realistic isochrone elevations along the transects analysed in this work are highlighted by thick lines $(5, 6\% K^{-1})$ B) Green and light blue lines depict the mean 0-9 ka and 9-38 ka accumulation along transect DC-X45afrom Cavitte et al. (2018) compared to the simulated accumulation for the same time with a precipitation forcing with $6\% K^{-1}$ temperature-precipitation scaling (dashed lines). The semi-transparent dashed lines depict the precipitation after accounting for the difference between the present-day reference data from van Wessem et al. (2014) and the observed modern surface mass balance estimate at EDC, which is $\approx 2.5$ cm/a (Stenni et al., 2016).

However, it is relatively small (4.8% vs. 11.4% RMSD) in comparison to the difference between pal (4.8% RMSD) and pd (20 − 56% RMSD) despite the fact that surface elevations for pd-pal and pd are very similar. All isochrone elevations simulated

in the pd case are unrealistic but also show a substantial spread for the parameter range tested here (see Section 2.2). Increasing either the basal friction (via the till friction angle) or the parameter controlling the sliding (via $q$) shifts the isochrone elevation by almost a third of the local ice thickness. However, even for parameter sets which lead to a growing ice sheet under present-day climate conditions (corresponding with an ice-sheet model parameterisation which leads to high basal drag and slower vertical and horizontal ice advection), the simulated elevation of the 96 ka isochrone is well below the observed one. This

shows that it is only possible to simulate realistic isochrone elevations, while achieving an overall ice sheet shape in agreement with present-day observations, if one takes into account the paleo-evolution of the ice sheet.

Tuning an ISM to present-day observations and mass-balance trends (pd) might yield a suitable fit to ice-sheet surface observations but fails to capture the integrated internal layering which is a product of the paleo surface-mass-balance as well as of paleo ice-dynamics. This might have important repercussions for paleo ice-sheet studies as well as for projections of

the future sea-level contribution of the Antarctic Ice Sheet, as the ice-sheet's initial state can affect its future behaviour over centennial or even decadal time scales (Seroussi et al., 2019) due to misrepresentations for example in the parameterisation of basal friction, the internal flow fields and the thermal state as a consequence of overfitting. Ideally, to assess the impact of

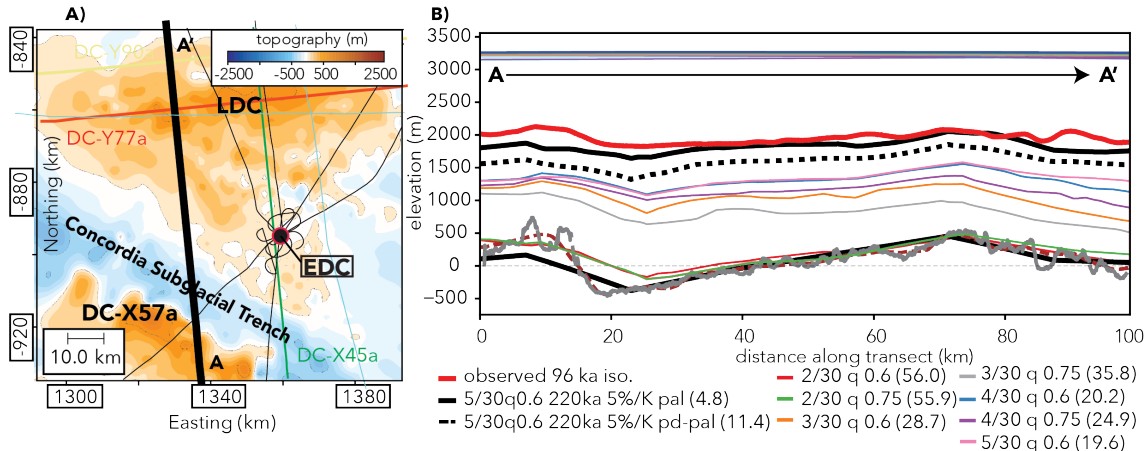

**Figure 7.** Comparison of the observed and modelled 96 ka isochrone along radar profile DC-X57a (red line in right panel) in the Dome C region: A) depicts the study area and the location of the radar transect corresponding to the simulated and observed isochrone. B) 96 ka observed isochrone elevation in red, paleoclimate simulation with $5\%\text{K}^{-1}$ precipitation scaling black line, present-day output of the paleoclimate simulation (black dashed line) and present-day equilibrium simulations as well as corresponding surface elevations (colored lines). The observed surface elevation is plotted in dark blue. Root mean square difference (%) between observed and modelled isochrone elevation is given by the numbers in the legend of B). Bedrock elevation in the pal model run is plotted in black and bedrock elevation derived from the radar transect in grey. The gridded 1km BedMachine bedrock elevation (Morlighem et al., 2020) is plotted for comparison as well (smooth dashed red line).

isochrone elevation calibration on ice-sheet-model projections, one would compare three cases where the model parameters are tuned a) via thickness/surface velocity inversion, b) against present-day surface observables and paleo-proxies (ideally

including a paleo spinup) and c) isochrone elevation calibration combined with b). However, this is work in progress and beyond the scope of this manuscript.

### 3.3 Impact of lower boundary conditions on simulated isochrone elevation (iii)

Next to the uncertainties associated with paleoclimate forcing fields and ISM parameterisations, the uncertainties in the basal boundary conditions (iii) applied in a large-scale paleo-ISM further complicate the computation of isochrone elevations. Areas

with sparse radar observations may have bed elevation estimates that differ from high-resolution radar data by several hundred metres (Karlsson et al., 2018; Morlighem et al., 2020). This can affect the basal flow regime and corresponding thermal state of the local ice column significantly. We would expect that line transects characterised by a bedrock profile in relatively good agreement with reality should yield a good agreement between modelled and observed isochrones as well. While this is true (e.g. for some areas of transect DC-X57a in the Dome C region, see Figure 9), the assumption cannot be generalised to all

East Antarctic transects discussed in this work as uncertainties in the parameterisations of ice flow (see Figure 7), paleoclimate forcing (Figure 5) and geothermal heat flux (Figure 8) can have a dominating or confounding effect over differences in the

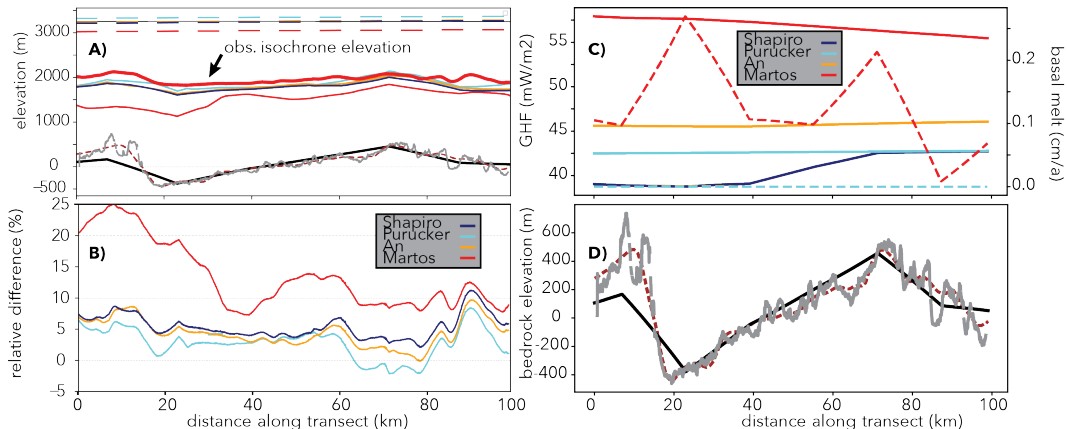

**Figure 8.** Illustration of model data along DC-X57a transect with a temperature precipitation scaling of $3\%\mathrm{K}^{-1}$. Panel A) shows the modelled surface (dashed lines, observed ice surface is depicted by the thin black line) and isochrone elevation under geothermal heat flux boundary conditions from Shapiro and Ritzwoller (2004); Purucker (2013); An et al. (2015); Martos et al. (2017). Panel B) depicts the isochrone mismatch (normalised by local ice thickness) between modelled and observed isochrone elevations. Panel C) shows the geothermal heat flux along the transect and the associated modelled basal melt (dashed lines). Panel D) highlights the bedrock elevation from the model simulation based on the Bedmap2 (Fretwell et al., 2013) at 16 km resolution (black line), bedrock radar reflection (Cavitte et al., 2016) (gray line) and Bedmachine Antarctica (Morlighem et al., 2020) data at 1 km resolution (red dashed line).

observed and modelled bedrock relief. Additionally, the impact of bedrock undulations on isochrone elevation will be stronger close to bedrock and might not be observable in relatively shallow isochrones where the surface mass balance is the dominating factor.

A further challenge complicating the comparison between simulated and observed isochrone elevations is the largely unknown distribution of geothermal heat at the base of the ice which is only inferred indirectly and exhibits large differences between datasets from Shapiro and Ritzwoller (2004); Purucker (2013); An et al. (2015); Martos et al. (2017). Comparison with simulations employing other datasets shows that the modelled isochrones fit well for geothermal heat flux from Purucker (2013) and An et al. (2015) but fail to reproduce the approximate isochrone elevation for the high geothermal heat flux from

Martos et al. (2017). The Martos et al. (2017) dataset produces widespread basal melting beyond (1 mm/a) in the region (see Figures 8 and 4 C). Geothermal heat flux is uncertain for most parts of the Antarctic Ice Sheet and regional differences between published datasets (Figure 8) can be substantial (Burton-Johnson et al., 2020; Talalay et al., 2020). Local basal melting occurs under forcing with the Shapiro and Ritzwoller (2004) dataset as well, but is largely limited to (0-1 mm/a) and lies between 0.25-0.5 mm/a around EDC. The computed basal melt rates with geothermal heat flux from Shapiro and Ritzwoller (2004)

and a temperature precipitation scaling of $6\%\mathrm{K}^{-1}$ are similar to the ones in Passalacqua et al. (2017) which are empirically determined to fit the EPICA Dome C ice core age scale. A more complete assessment of the relative influence of basal and surface boundary conditions as well as model parameterisations on East Antarctic isochrone elevations across the whole ice

column is beyond the scope of this study but could be the focus of an upcoming paleo-ISM intercomparison.

## 3.4 Caveats to modelling isochrones with large scale ISMs

Differences between modelled isochrones (based on paleo-velocity fields) and their observed elevation above bedrock are generally small for all isochrones sampled in this study that are younger than the LIG ($< 120$ ka, root mean square error of RMSD $< 5\%$). It is striking, however that for isochrones older than 120 ka the gap between model results and observation widens (see Figure 9). Due to the lack of climate model data for glacials and interglacials preceding the LIG, we estimated the climate forcing before 130 ka BP to be a linear combination between the climate state of the LGM and the LIG using the EPICA Dome C deuterium record as an index to interpolate between the two climate states (see section 2.1 in Sutter et al., 2019). Naturally, this approach is just an approximation of the actual transient climate conditions and therefore will necessarily lead to discrepancies in the surface mass balance as well as the ocean forcing. Another potential cause of the relatively poor representation of isochrones before 120 ka BP is the high circum-antarctic subsurface ocean temperature peak assumed for the LIG (ca. 2°C). Due to coastal thinning and potentially grounding-line retreat caused by elevated ocean temperatures, this might lead to an exaggerated ice flow in the major drainage basins such as the Wilkes Basin (see Fig 10) or the Aurora Basin, in turn potentially affecting even remote regions such as Dome C. In fact, the model-data-observations mismatch is especially poor for the sections of the Dome C transects close to the Aurora Subglacial Basin (see Figure 9 B) which could also point to an issue with the parameterisation of basal drag in subglacial basins (see section 4).

## 4 Large scale modelling of the internal architecture of the East Antarctic Ice Sheet

In the following, we expand our view to the whole of the East Antarctic Ice Sheet, assessing how valid the choices of model parameters and the paleo-accumulation forcing derived from Dome C are when it comes to simulating the evolution of the ice sheet. The focus on the EDC ice core region in the previous section allowed us to appreciate the impact of uncertain forcing fields and bedrock elevation on a relatively small length scale (100 km). We now turn to larger distances, where the Dome C-derived transient paleoclimate forcing and model optimisation might lead to more pronounced divergence between the modelled isochrone elevations and the radar data. We first assess the 90 ka isochrone along transect CEA-10 from Winter et al. (2019) (see Figure 10). It starts close to Talos Dome crossing the upper reaches of the Wilkes Subglacial Basin after which it traverses the Concordia Subglacial Trench, passing EDC and LDC and finally arching upwards, ending at Lake Vostok (total distance ca. 1500 km). In the second part of this section we discuss the isochrone geometry between Dome Fuji and the EPICA Dronning Maud Land ice core (EDML).

### 4.1 Simulated isochrone elevation along the Talos Dome - Lake Vostok transect.

Figure 10 illustrates nicely that there are two main regimes covered by profile CEA-10. The northern half of the transect close to Talos Dome (bottom left in Figure 10) is dominated by the imprint of the Wilkes Subglacial Basin with a large misfit between

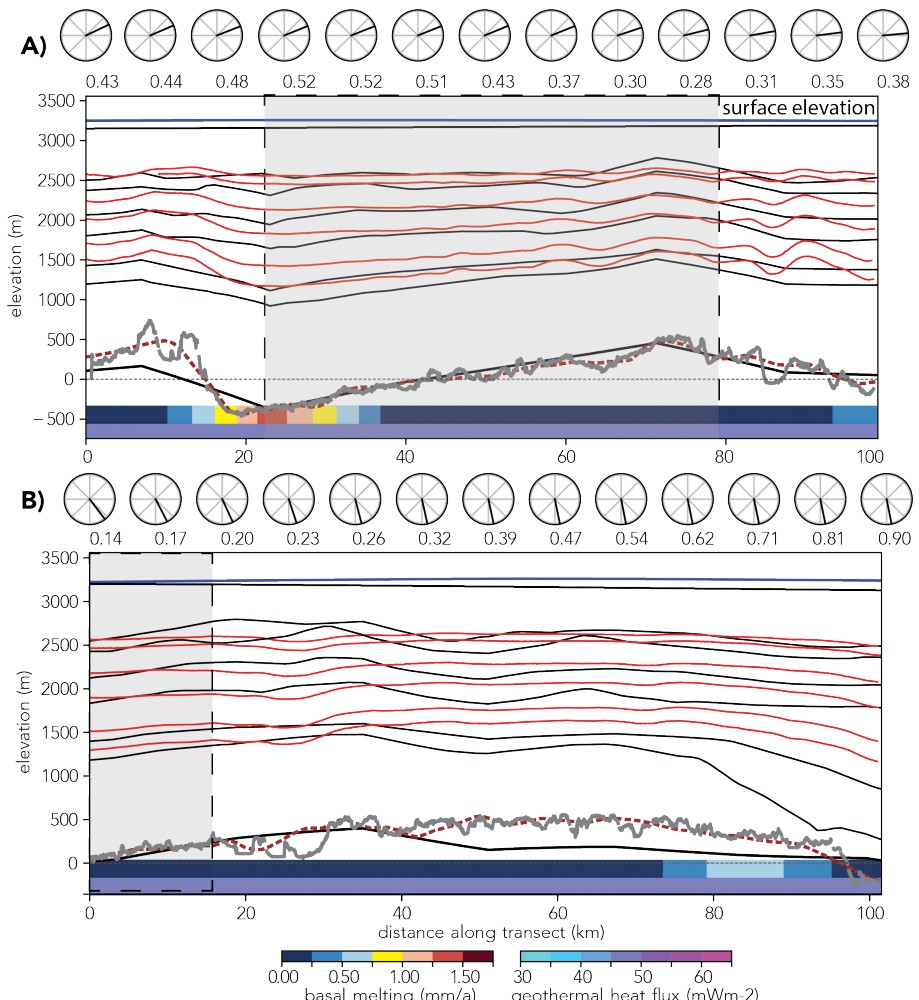

**Figure 9.** Modelled (black lines) and observed isochrones (red lines 38, 48, 74, 96, 130, 160 ka from top to bottom) for transect DC-X57a (A) and DC-Y77a (B) (orientation from left to right as depicted in Figure 7 A). The grey vertical bars highlight areas with low elevation differences in the low resolution model bedrock input (black line) compared to the high resolution radar observations (grey line) which generally correspond to a good match of modelled and observed isochrones. Bedrock and surface elevation in the model runs is plotted in black. Observed surface elevation is plotted in dark blue. The 1 km BedMachine data (Morlighem et al., 2020) is depicted by the red dashed line. The labeled circles denote the surface flow and magnitude in m/a relative to the direction of the transect. The coloured bars on the bottom depict the geothermal heat flux (Shapiro and Ritzwoller, 2004) and the computed basal melt rate along the transect.

the modelled and observed isochrone ($\approx 26\%$ RMSD along km 0-500 in Figure 10, compared to $\approx 8.3\%$ and $\approx 4.8\%$ along km 500-1000 and 1000-1500, respectively). The striking excursions of the modelled isochrone elevation in Figure 10 point towards an internal vertical ice motion that is too fast. This could be due to the general heuristics involved in determining the basal friction in subglacial basins (low friction in deep basins). Furthermore, past elevated ice flow episodes of the marine

Wilkes Basin Ice Sheet caused by thinning of the coastal ice cover for example during interglacials (Sutter et al., 2020) could have mediated drawdown of isochrone elevation. While a spatially varying temperature-precipitation relationship could also affect the isochrone elevation in this region, this seems not to be the case here. We show this by computing the 90-ka isochrone elevation in a paleo-simulation with a temperature-precipitation relationship of $8\%K^{-1}$. This scenario cannot mitigate the drop in elevation in the first 300 km of the transect and exacerbates the deviation between simulated and observed isochrone elevation along km 500-1500 ($\approx 16.7\%$ RMSD along km 0-500 in Figure 10, compared to $\approx 8.7\%$ and $\approx 7.8\%$ along km 500-1000 and 1000-1500, respectively). We did not test temperature precipitation relationships higher than $8\%K^{-1}$, as they would be beyond what is reconstructed from East Antarctic ice cores. Geothermal heat flux can also be ruled out as for the most part the thermal regime in the dataset from Shapiro and Ritzwoller (2004) at Dome C is similar or even lower. On the basis of the boundary conditions used here, we suggest that the main reason for the isochrone mismatch lies in the parameterisation of basal drag and ice flow. The basal friction in the model is a function of bedrock elevation (see section 2.2) and decreases with depth, therefore it is lower in bedrock depressions especially below sea level where sliding is thus higher. Also, the relatively coarse resolution (16 km) used here leads to a "smoothed-out" bedrock profile (see Figure 8) potentially favouring faster basal ice flow. A "rougher" bedrock profile might impede sliding in drainage regions. Another aspect could be the choice of the so called "enhancement factor" in the shallow ice approximation $SIA_e$ which is a crude tuning parameter to accommodate for anisotropic ice. Here, $SIA_e = 1.0$ which is low compared to values used in other studies (generally, a lower value of $SIA_e$ leads to slower ice flow) so we do not expect that the enhancement factor plays a dominant role. Modelled isochrone elevations in the present-day equilibrium ensemble (case pd) are far off the observed target elevation throughout the CEA-10 transect (see Figure 11). This finding is in agreement with the Dome C region pd-case (section 3.2).

## 4.2   Simulated isochrone elevation along the Dome Fuji - Dronning Maud Land transect.

We now focus on the 74 ka isochrone connecting Dome Fuji ($77°32'S$, $38°70'E$) and the EDML ice core ($75°0'S$, $0°068'E$). Figure 12 illustrates the topographic characteristics of the area enclosing Dome Fuji and EDML. In contrast to Dome C, which is surrounded by bedrock below sea level in the outskirts of the Aurora Subglacial Basin, the ice bed at and around Dome Fuji is elevated above sea level for several hundreds of kilometres in every direction. This means that the yield stress computed by the ISM will be very high in a large area, making the impact of sliding on internal ice velocities negligible. Both transects DML-VIII23 and DML-IV24 are characterised by mountainous subglacial terrain above sea level (see Figure 12 A). Any mismatches between the observed and modelled isochrone elevations is most probably due to uncertainties in the climate forcing or computed basal melt patterns. Basal melting at the bed of the ice along the radar tracks is largely unknown, as is the temporal paleo-accumulation pattern. A direct comparison to proxy or observational data is therefore not feasible. In the future observation-based estimates of the presence or absence of basal melt (e.g. Fujita et al., 2012; Passalacqua et al., 2017; Karlsson et al., 2018) could be utilised to locally attribute mismatches between modelled and observed isochrone elevations to inconsistencies in basal melting between model and observations.

We limit ourselves to a qualitative discussion of the model-data mismatch trying to identify persistent patterns. Looking at DML-VIII23, the modelled isochrone elevation fits well with the observed radar data (RMSD $< 5\%$ for the whole transect) and

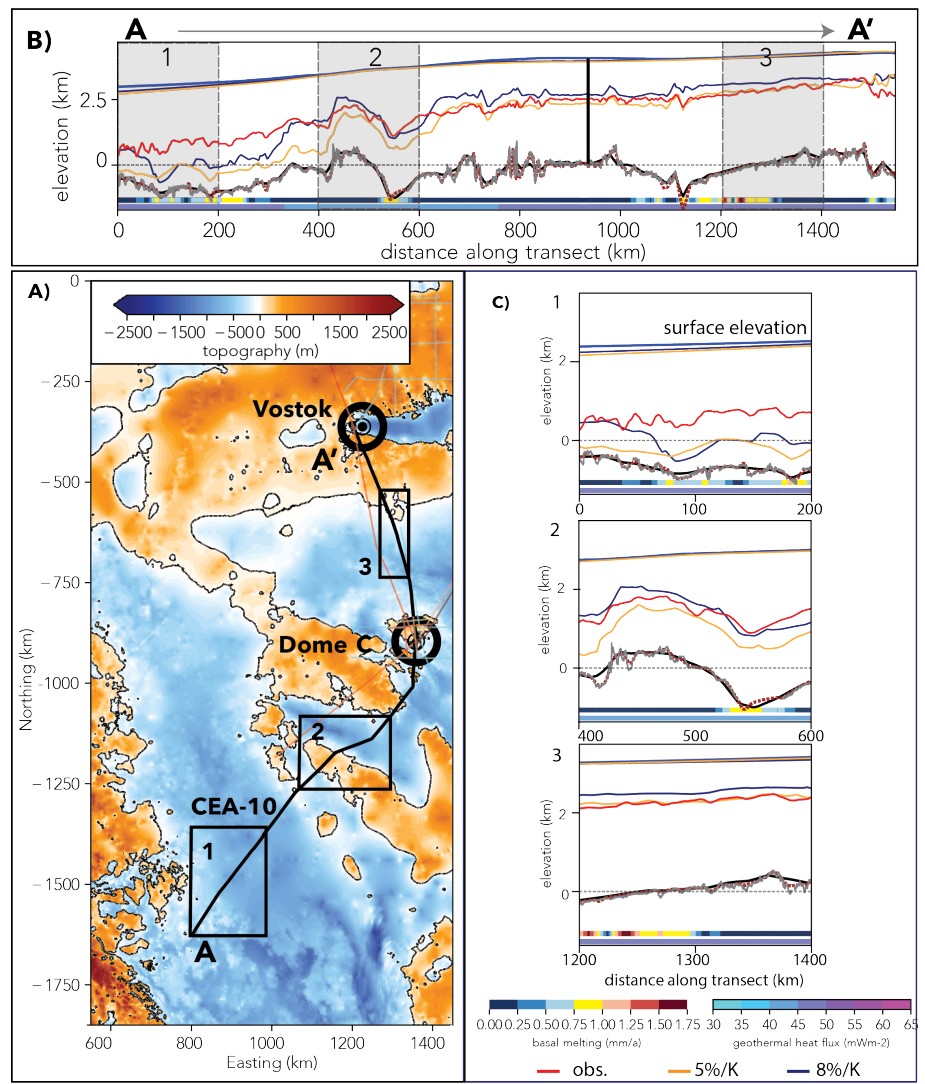

**Figure 10.** Overview of transect CEA-10. Panel A) shows the bedrock elevation from Morlighem et al. (2020) at 1 km resolution overlain by the CEA-10 transect in black and other radar transects analysed in this study (thin coloured lines). Panels B) and C) illustrate the ice surface (dark blue observed, modelled - blue $8\%\text{K}^{-1}$, yellow $5\%\text{K}^{-1}$), bedrock and isochrone elevation of the 90 ka radar reflector (modelled - blue $8\%\text{K}^{-1}$, yellow $5\%\text{K}^{-1}$, observed - red) along transect CEA-10 (The position of the EDC ice core is illustrated by the vertical black line in panel B). Bedrock elevation in the pal model run is plotted in black and the BedMachine Antarctica 1 km-grid resolution bedrock in red. Shaded areas in B are shown additionally in a detailed view (C). Sea level (0-elevation) is depicted by the black dotted line. The coloured bars on the bottom of B) and C) depict the geothermal heat flux and the computed basal melt rate along the transect.

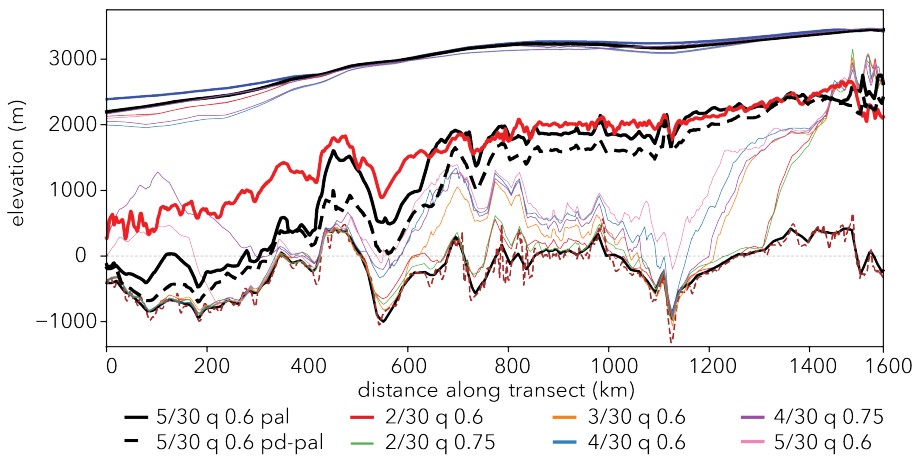

**Figure 11.** Simulated and observed isochrone elevations along transect CEA-10. The thick red line depicts the observed 90 ka radar reflector, the black line the 220 ka paleo-simulation with a paleo-precipitation-temperature relationship of $5\%K^{-1}$. The dashed black line shows the isochrone elevation computed with the respective present-day time slice (case pd-pal) of the paleo simulation (pal). The thin coloured lines represent the simulated isochrone and surface elevation from the present-day equilibrium ensemble (pd). Bedrock elevation in the pal model run is plotted in black and bedrock elevation derived from Morlighem et al. (2020) in red. The observed surface elevation is plotted in dark blue.

is of similar accuracy as for the 100 km Dome C data for the same transect age (RMSD $3.5 - 5.0\%$). In contrast to the lower half of CEA-10 (previous section), which is situated above a deep bedrock depression associated with low basal friction in the model, the whole transect DML-VIII23 is characterised by relatively flat bedrock above sea level. The comparison between

445 DML-VIII23 and CEA-10 potentially highlights a methodological deficiency (leading to unrealistic internal flow and basal sliding) as the isochrone mismatch in CEA-10 cannot be remedied by a surface mass balance correction.

Modelled isochrone elevations based on simulations employing inversion techniques to determine basal friction and/or a paleo-ensemble covering a larger parameter spread would be necessary to assess whether this mismatch can be further reduced using state of the art models today.

In case of DML-IV24 a slightly different picture emerges. For the first half of the transect, the modelled isochrone is in very good agreement to the observed elevation data (except in areas with strong deviations between the 16 km BedMap topography and the high resolution radar observation), but the second half of the transect shows large deviations in the isochrone elevation leading to an overall RMSD of ca. $10\%$ for the whole transect. In this case, it is not straightforward to identify the causes for the mismatch as the bedrock is above sea level along the whole transect and there is no clear correlation between elevated

basal melt rates and dips in modelled isochrone elevation. One potential reason could be a relationship between temperature and precipitation anomalies in the EDML region which strongly differs from the continental-scale temperature precipitation scaling of ca. $5 \pm 1\%K^{-1}$. This would affect the surface mass balance forcing and therefore the elevation of the isochrone. However, the proxy-based paleo-precipitation-temperature relationship at EDML is very similar to the continental mean albeit

with considerable uncertainties of $\pm 2.8\%$ for EDML (Frieler et al., 2015). In fact, using the ISM results of a simulation with a precipitation-temperature relationship of $8\%\mathrm{K}^{-1}$, which is at the upper end of reconstructions for EDML, the match of the computed isochrone elevation close to EDML is improved considerably (see Figure 12). Thus, the assumption of a simple and spatially uniform accumulation/temperature scaling may be valid on the high plateau of the East Antarctic Sheet but not necessarily hold for areas closer to the coast, where synoptic activity can dominate the spatial and temporal variability in precipitation (Welker et al., 2014). This could only be remedied in coupled atmosphere/ice-sheet model runs that are able to resolve such synoptic activity. For computational reasons, however, this is currently not possible for such long-term runs as performed in this study. One potential solution would be to use a spatially heterogenous accumulation/temperature scaling informed by both coastal and interior ice core reconstructions.

## 5  Conclusions

We present the first attempt to constrain Antarctic paleoclimate forcing and parameterisations of ice flow in a continental-scale ice-sheet model by comparison of simulated englacial layers against a pool of observed Antarctic isochrones.

– We are able to reconstruct most large-scale englacial layer features of the observed isochrones and show that it is possible to simulate the observed internal structure of the Antarctic Ice Sheet even at coarse resolution. We identify mismatches between modelled and observed isochrone elevations, which can be traced back to the transient paleoclimate forcing employed in our model runs that makes use of a linear paleo-temperature-precipitation relationship. The forcing is derived from ice core reconstructions in combination with paleoclimate model data. This does not take into account the spatial heterogeneity of paleo temperature-precipitation relationships and effects of synoptic precipitation variability. Our isochrone modelling efforts motivate the use of a regionally-refined temperature-precipitation scaling to improve paleo ice-sheet simulations and consequently the paleo spinup for model-based future projections.

– We further show that our computed isochrone elevations are in very good agreement (within $\pm 5\%$ RMSD of the local ice thickness) with observations in slow flowing regions such as at or near ice divides, especially at locations where the ice-core based proxy reconstruction of the temperature-precipitation scaling matches the one used in the forcing. However, we find key isochrone elevation mismatches in the proximity of, or above, marine glaciated areas of the Antarctic Ice Sheet, which points towards a poorly constrained parameterisation of basal drag. There seems to be a systematic overestimation of vertical advection over subglacial basins and in drainage sectors. This is most probably a commonality of all ISM setups using similar heuristics for basal friction as we use here; i.e., the majority of ISMs used for paleo-studies. This raises the question whether paleo-simulations and projections of ice dynamics in subglacial basins might be subject to a systematic bias. We show that using englacial layers as a tuning target in ice-sheet modelling is ideally suited to identify such biases, as surface observables such as ice-sheet elevation do not necessarily reveal inconsistencies in the subsurface flow field. Expanding the existing radar observations into regions of dynamic ice flow away from ice divides and over subglacial basins would provide invaluable constraints for paleo-ISM simulations.

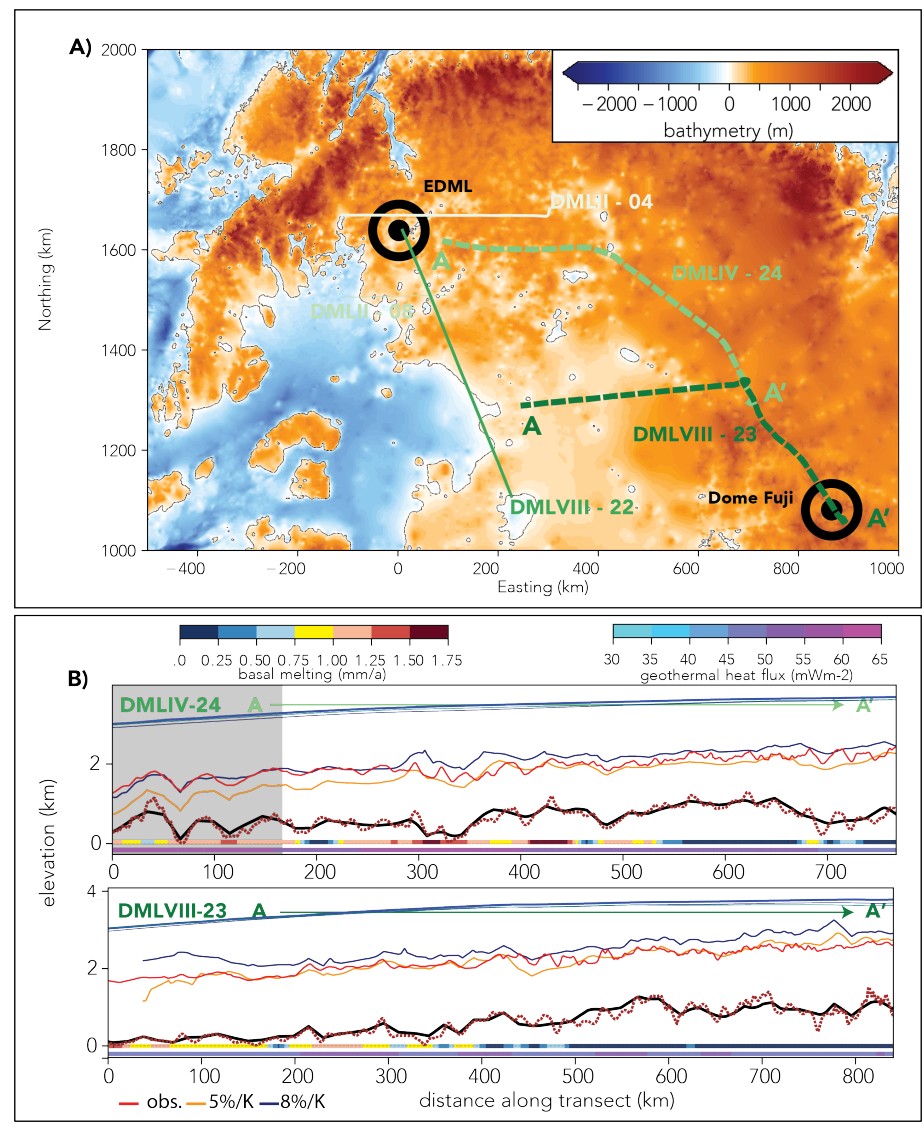

**Figure 12.** Overview of transect DML VIII-23 and IV-24. Panel A) shows the bedrock elevation from Morlighem et al. (2020) at 1 km resolution overlain by the DML transects in green and other radar transects analysed in this study (thin coloured lines). Panel B) illustrates the ice surface, bedrock and isochrone elevation of the 74 ka radar reflector (modelled - blue $8\%\mathrm{K}^{-1}$, yellow $5\%\mathrm{K}^{-1}$, observed - red). The model bedrock elevation is depicted in black while the original radar data and the BedMachine Antarctica 1 km-grid is plotted in red. The coloured bars on the bottom of B) depict the geothermal heat flux and the computed basal melt rate along the transect.

- The reproducibility of isochrones dramatically increases if the ice sheet has been simulated transiently over the last several glacial-interglacial cycles. In contrast, flow fields computed in equilibrium simulations under constant present-day forcing produce isochrone elevation mismatches of up to two-thirds of the local ice thickness. Varying the parameter space relevant for basal sliding for a set of parameters that produce an equilibrium sea-level equivalent ice volume of the Antarctic Ice Sheet within $\pm 2$ m of present-day observations cannot remedy this mismatch. This points towards critical misrepresentations of the ice sheet's internal flow for model setups where only present-day climate forcing is taken into account and the model is solely tuned against present-day 2D and 1D observables. Even small biases (e.g. due to overfitting against uncertain input fields) in the initial model state can impact ice sheet dynamics and therefore estimates of future ice sheet sea level contributions. We make the case that the paleo-evolution of an ice sheet should be considered both for reconstructions as well as projections of ice sheet changes and that isochrones are ideally suited for this purpose.

- When using isochrones as a tuning target for paleo-ISMs, two key uncertainties prove difficult to account for: 1) geothermal heat flux fields remain poorly constrained (new, e.g. Stal et al. (2021), and upcoming datasets might reduce this uncertainty) and can have a strong influence on isochrone elevations. 2) Uncertain bedrock elevation in regions with gaps in radar surveys affect modelled isochrone elevations especially for isochrones close to bedrock. However, for areas covered by high resolution radar transects, this aspect can be quantified by comparison to the model bedrock elevation. Combining isochrone elevations, present-day observables and paleo proxy data in the calibration of ice sheet model setups helps to mitigate aforementioned uncertainties and prevent overfitting.

- A model intercomparison investigating isochrone elevations based on a variety of model physics and forcings could shed light on systematic misrepresentations of ice flow and, thus, internal stratigraphy in current generation ISMs. For example, the impact of different calibrations of basal drag on modelled isochrone elevations, such as inversion methods based on surface elevation and ice flow, could be elucidated in such an intercomparison. Our post-processing approach would allow for such an intercomparison as it forgoes the need to implement a Lagrangian tracer module into the respective ISM. We make the case that the internal stratigraphy of the Antarctic Ice Sheet can serve as a valuable data benchmark for continental ice-sheet modelling as it provides a three dimensional tuning target which is imprinted with the complete climate and flow history of the ice sheet.

We conclude that this approach should be used alongside traditionally employed tuning targets such as ice volume, surface velocity or grounding-line positions. While analysing the match of an ISM simulation with the internal stratigraphy is not as straight forward as using surface observables, it could improve both paleo ice-sheet reconstructions as well as sea-level projections due to more realistic initial ice-sheet configurations. Efforts such as AntArchitecture's to provide a compilation of all observed englacial layers will provide an invaluable data benchmark for future ice-sheet modelling efforts. Looking ahead, it would be desirable to develop a standard protocol to tune ISMs against the internal stratigraphy of the Antarctic Ice Sheet. This would facilitate the evaluation of a new generation of model simulations which are constrained by the climate and ice dynamic memory encapsulated within the ice.

*Code and data availability.* Both PISM and OceanParcels are open source and available via https://pism-docs.org/wiki/doku.php and http://oceanparcels.org/index.html, respectively. The isochrones from Winter et al. (2019) are available via Pangaea https://doi.pangaea.de/10.1594/PANGAEA.895528, all model data is available upon request.

*Video supplement.* Tracer migration from 90-0 ka BP. Tracers are seeded at the surface at 90 ka BP and transported with the ice flow over time. Colouring illustrates absolute tracer elevation. Available at https://doi.org/10.5446/50125

*Author contributions.* JS devised the experiments and ran the simulations. JS, HF and OE carried out the analysis. JS wrote the manuscript with contributions from HF and OE.

*Competing interests.* OE is CEIC of The Cryosphere

*Acknowledgements.* We would like to thank Marie Cavitte, Gwendolyn Leysinger Vieli, Anna Winter, Lucas Beem and David Ashmore for providing the isochrone data and helpful discussions as well as Philippe Delandmeter and Victor Onink for their support with setting up OceanParcels for the use with PISM-data. JS is grateful for funding from the Deutsche Forschungsgemeinschaft under personal grant SU 1166/1-1 (Project WANT-Ice). Hubertus Fischer gratefully acknowledges the long-term support by the Swiss National Science Foundation (SNSF). Development of PISM is supported by NSF grants PLR-1603799 and PLR-1644277 and NASA grant NNX17AG65G.

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
