# Peer review of "Investigating the internal structure of the Antarctic Ice Sheet: the utility of isochrones for spatiotemporal ice-sheet model calibration"

_The Cryosphere, 2020_

## Referee Comment (RC1) · Anonymous Referee #1 · 15 Jan 2021

The manuscript by Sutter et al. (2020) shows how the ice sheet internal layer structure can be exploited to understand and diagnose ice sheet model output. The authors present a clear case for the utility of comparing isochrones derived from observations and ice sheet model simulations to determine ice sheet model performance, particularly highlighting where we need:

1. Better constraints on boundary conditions (e.g. bed topography; geothermal heat flux);

2. Better constraints on climate forcings (e.g. spatial variation in paleo accumulation rates);

[Figure]

3. Better constraints on ice sheet model parameters (e.g. basal drag over marine sectors of the ice sheet);

4. Long term simulations to adequately represent 3D flow fields and ice sheet geometries.

The diagnostic method presented in this manuscript (i.e. use of particle tracer method) is freely available and can be readily applied to any ice sheet model output, making this diagnostic tool accessible for ice sheet modellers.

The manuscript addresses a highly relevant scientific question, especially with the work of the AntArchitecture project. To the best of my knowledge the concept is novel, and the scope of the model simulations and comparison with observations is appropriate to support the interpretations and conclusions, and to demonstrate broad applicability of the method to the ice sheet modelling community. Overall, this is a worthwhile study that is certainly within the scope of TC.

I have a few main comments that should be addressed before publication, mostly related to finessing the structure and flow of the manuscript for improved readability, and on how generalizable the results are for the whole ice sheet. I also have included a list of specific comments that should be addressed.

**Main comments**

1. The title

   The title should be tightened to be more specific. E.g. "Simulating the internal structure" could be qualified, especially given that the isochrones were derived from simulated velocity and geometry fields, rather than calculated online in the ice sheet model. Suggest something like: "Investigating the internal structure of the Antarctic Ice Sheet: the utility of isochrones for spatio-temporal ice sheet model calibration" OR "On the use of isochrones as a novel diagnostic for ice sheet model performance"

2. Structure and readability of sections 3-5:

I found the structure and flow of sections 3-5 difficult to follow, and because of this the findings of the study – particularly with respect to clearly identifying the sources of mismatches between observed and simulated isochrone depths – and their significance were weakened. For example, L195-206 introduces the main goal and three main potential sources of model-obs mismatches. While the sections that follow contain discussion of each of these sources, it is not always clear which source is being considered (e.g. with subsection headers), and there is some repetition (particularly sections 3.2 and 4). It'd be good to see a restructuring through sections 3 and 4, with distinct subsections to systematically address each source of mismatch, first with respect to Dome C (section 3), then with respect to the broader EAIS (section 4).

In each section, the aim should be named early in the paragraph so that it's clear why certain experiments are being assessed, and the significance of the results. E.g. for L253-276, evaluating the model parameterisation, the aim of that paragraph is stated on L262-263, but should be moved to the start. This would help in interpreting the results.

In section 4 on P16 L341-376, the presentation of the results/physical conditions is mixed up with the discussion of the processes that could contribute to obs-model mismatches and how they contrast regionally. It'd be good to first clearly discuss the Dome C and Dome Fuji results. E.g. "for DML-VIII23, the bedrock elevation is above sea level and relatively flat. For this transect, the obs-model mismatch..." Then in a new paragraph, contrast the obs-model mismatch between each of the transects (DML-VIII23, CEA-10, DML-IV24) to highlight the sources/processes that could contribute to the mismatch. It'll then be easier to come away with a clear picture of what is causing what on the larger scale.

It'd be helpful for readability of the conclusions section to restructure into 4 paragraphs:

- Summarise how well the paleo model outputs match the main observed features of the internal ice sheet structure and capture broad-scale SMB patterns.
- Summarise how the method helps to understand the processes under-lying/sources of mismatches between observed and simulated isochrone depths, with specific reference to the four areas identified in comment 1. above: (a) boundary conditions; (b) forcing time series; (c) initial conditions/parameters; and (d) simulation time.
- Summarise where more efforts are needed: (a) constraints on the spatial variability in paleo accumulation rates; (b) constraints on basal drag in ma-rine sectors.
- Discuss the model intercomparison

3. Reporting of basal drag and methodology:

   One of the main findings is that the basal drag can play a large role in the mis-match between observed and simulated isochrones. I.e., the large mismatch in some regions is due to an overestimation of vertical velocities where there is low basal friction. However, the actual basal drag is not reported in this paper for the simulations. This makes it very difficult to ascertain whether the aims of the study that relate to basal drag have been adequately addressed, and the degree to which the discussion around the accuracy and constraints on the basal drag applies. The basal drag coefficient should be reported for each of the simulations as a separate figure in the main body of the manuscript. It'd also be good to see the drag coefficient as a separate panel on figure 9. I recommend adding sup-plementary material that describes how the basal drag was calculated for each of the simulations.

   It'd also be worth reiterating some of the salient features of the model setup/methodology, particularly those aspects that are relied upon for interpre-tation of the results. For example, how was the model tuned at Dome C? This is

key in understanding point (i): the impact of the paleoclimate climate forcing on the model-obs mismatch (see specific point below).

4. Generalisation of results to the large-scale:

   I'm slightly concerned that the abstract (and then the conclusions) oversells the results by use of some general terms e.g. "Antarctic Ice Sheet", "the interior", and "subglacial basins". The results for the isochrones that are analysed certainly support the interpretations; however, in reality, these isochrones only cover very small parts of the East Antarctic Ice Sheet. How representative are the results for these particular observational isochrones for the large-scale ice sheet? Please clarify this, and be more specific in the abstract and conclusion to reflect this. Please also consider discussing the generalisability of the results in more detail in section 4.

5. Grammar and spelling:

   I have noted some grammatical and spelling points below, but not all of them – others should be able to be corrected fairly easily with a standard LaTeX spelling checker or online e.g. grammarly.com. Some of the sentences are also long and a little bit wordy – it would be worth shortening the longer sentences for improved readability.

**Specific comments**

P1 L6. "We simulate observed isochrone elevations within the AIS via passive Lagrangian tracers" >> "We calculate isochrone elevations from simulated AIS geometries and velocities via passive Lagrangian tracers".

P2 L31. Check citation style. Also on P6 L152-153, P12 L283, P18 L367.

P2 L45. What does "in scope of" mean here? "Relevant to"?

Table 1. I couldn't see the analysis of a number of these isochrones (e.g. CEA-7,8,12,13). Were these reported within the figures and text?

P6 L134. Sentences starting from "We use" to a new paragraph.

P6 L136-139. The sentence starting with "Based on the analysis" was a bit confusing. I took it to mean the following: (a) that the observed isochrone data used in this study (derived from Winter et al. 2017) are assumed to have a maximum age uncertainty of 1 ka; (b) that the observed isochrone data are all above or at 2000 m depth; (c) that the age uncertainty nonlinearly increases with depth; (d) that the age uncertainty of the observed isochrone data is always lower than the uncertainty in the simulated data. Is this what is meant? This sentence could be reworded for improved clarity.

P6 L138. "2/3 or" $>>$ "2/3 of"

P6 L154. "relatively coarse resolution" $>>$ "relatively coarse resolution model grid"? Does the mesh size evolve over time with grounding line migration or is it static? Some more details on the model experiments would be helpful here.

P8 L175. How is "accurate enough" determined? How do we know that the misfits between the radar isochrones and the simulated isochrones using this method and the ISM are not sensitive to the temporal and spatial resolution of the ISM output and the Lagrangian particle tracking algorithm? It would be good to see a sensitivity analysis or uncertainty quantification here.

P8 L184 "We also tested other seeding strategies..." What was the outcome of these tests (i.e. was there any sensitivity to seeding mask)? A section quantifying the uncertainty in the tracer method would be appropriate in a supplementary material document.

P8 L201. "the model ensemble was tuned". What does this mean? Please provide specific details of what fields were tuned and to what data.

P8 L201-202. "To assess the impact of model" $>>$ "To assess the impact of (ii) model"

Figure 3. For this figure and for figures 4, 6, 8, and 9, please make the figure bigger (e.g. textwidth) and increase the font size. The blue lines in panel C are difficult to see - perhaps use black instead - and it'd be great to label the transects in panels B and C (see also comment on figure 4). In the caption, I wasn't sure what this sentence meant: "Due to the small mismatch… are discernible." Does this mean that the background colour is relatively uniform? Consider modifying the colour bar to zoom to the relevant colour range represented in the figure. Also, "strong" $>>$ "large" in the second last sentence of the caption.

P9 L214. "To evaluate the validity of the forcing approach in Sutter et al. (2019).." Earlier, it is mentioned that the model ensemble is tuned to match the regional configuration near Dome C. For what paleoclimate forcing was this tuning carried out? Please comment on whether/how this tuning might impact the capacity to assess the validity of the forcing approach.

Figure 4. The transect lines and labels are difficult to see on panel A - please make the lines thicker and more contrasting with the background. It's difficult to determine which line is which on panel B (the brown/purple/red colours are similar - perhaps choose more contrasting colours for the bed elevation). Please also make the lines thicker and include a legend on panel B?

P10 L219. "DC-57a" is not marked in bold in figure 4A (that's DC-X45a). Which transect is referred to here? Suggest labelling all transects more clearly on the figures.

P10 L221. Why use 3, 5, 6, and 8

P10 L225. "completely reproduces". What does this mean? The red line in panel B is sometimes outside of the 5-6

P10 L227. "at least for the interior of the East Antarctic Ice Sheet". The results show that is valid for Dome C (and given that the model is tuned for this region). It's not clear that this conclusion can be extended to the interior of the EAIS in general. Surely this

depends on the degree to which the climate forcing is appropriate in other regions? (which is indeed addressed in section 4).

P10 L230-231. This result might be related to the fact that the method employed in Martos et al. (2017) is not physically realistic. For a description and an updated GHF product for all of Antarctica, see: Stål, T, et al. "Antarctic geothermal heat flow model: Aq1." Geochemistry, Geophysics, Geosystems: e2020GC009428, https://doi.org/10.1029/2020GC009428.

P11 L242-252. I found this paragraph difficult to follow. First, the discussion of the 0.5 cm/a difference between observations (Stenni et al., 2016) and the models simulations + RACMO could be clearer. Is there a reason the simulations match RACMO but not obs? Second, for clarity please reference panel B in the text (e.g. "When we compare (figure 5B) our simplified...") and describe the bias correction that is used. Consider restructuring the paragraph for clarity, and add labels for the different curves in figure 5A.

P11 L251. "along ice divides" >> "along the ice divide near EPICA Dome C"

Figure 5. What are the semi transparent solid lines in panel A? Bias corrected values from the simulation? Please describe, along with the dashed lines, in the figure 5 caption.

P12 L253. "where" » "were"

P12 L256. Here 5

P12 L262-276. Should "pal", "pd-pal", and "pd-pd" be in italics here and through this paragraph?

P12 L267-276. This is a really neat and interesting result. It's also interesting because presumably unknown parameters (e.g. the basal friction coefficient) are also somewhat uncertain in the initialisation of the paleo simulation, but the pal run yields isochrone depths that are within a few

P12 L278-279. "are small" $>>$ "are generally small". E.g. the difference between obs and sim isochrones for 38 ka along DC-Y77 between ~15-25 km is much larger than for any other isochrone in this portion of the transect.

P12 L286. "antarctic" $>>$ "Antarctic". Here and elsewhere.

P13 L293. "60 and 90 ka" $>>$ "90 and 60 ka". How much uncertainty does the shorter spin-up time for 90 and 60 ka introduce?

P13 L297-298. A bit of a jumbled sentence. Suggest: "Areas with sparse radar observations may have bed elevation estimates that differ from high-res radar data by several hundred metres."

P13 L304. "differences in the observed and modelled bedrock relief" $>>$ "differences in the observed and modelled bedrock relief (e.g. see section XX)"

FIgure 7. Final line: "The coloured bars on the bottom of" $>>$ "The coloured bars at the bottom"

P14 L310. "focus of an upcoming paleo-ice-sheet model intercomparison". Great idea.

Figure 8. In the second last line of the caption, "normalised with" $>>$ "normalised by"

P15 L322. "The northern half of the transect bottom part in Figure 9". It's not hugely clear which segment of the transect this refers to - perhaps add demarcation on panels A and B of figure 9.

P16 L334. "The basal friction in the model is a function of bedrock elevation". Please provide the equation and description of the basal friction calculation, perhaps in supplementary material.

P16 L347. "isorchrone" $>>$ "isochrone"

P16 L349. "We limit ourselves to. . ." New paragraph.

P16 L356-358. Move "which encompasses...Wilkes Subglacial Basin" to earlier where

CEA-10 results are first introduced. Reword remaining part of sentence: "The comparison between DML-VIII23 and CEA-10 potentially highlights a methodological deficiency (leading to unrealistic internal flow and basal sliding) as the isochrone mismatch in CEA-10 cannot be remedied by a surface mass balance correction."

P16 L365. "Dome C". Should this be Dome F or Dome C here and also on page 18? I'm not sure I understand this argument if it's Dome C.

Figure 9. Consider a vertical line on panel B indicating the Dome C location so that we can compare GHF here to the northern portion of the transect. Consider also demarcating (e.g. with a coloured vertical line) the "northern portion/bottom half" of transect in panel B, and perhaps with two "x" the same region on panel A. For panel C, here and in figure 8, I suggest using a different colour map for the basal melt rate - the blue-white-red transition is usually used for differences (where white is 0). In the second line of the caption: "beige" looks more like dark red to me.

P18 L379-380. It would be better to reword as follows: "We are able to reconstruct most large-scale englacial layer features of the observed isochrones..."

P18 L394. "A model intercomparison". New paragraph

Figure 10. Should "DMLVIII 22" on panel A be "DMLVIII 23"? It's hard to see the dashed lines on panel B – please make the lines thicker.

P20 L405. "This would facilitate" $>>$ "This would facilitate the evaluation of"

---

## Referee Comment (RC2) · Anonymous Referee #2 · 4 Feb 2021

Summary: The authors present results from a study focused on utilizing the internal structure of the Antarctic Ice Sheet, as surveyed with ice-penetrating radar, to reveal and validate the paleo evolution of the ice sheet. This approach also has the potential to more thoroughly initialize an ice-sheet model used for future projections of mass loss and sea-level rise.

The new approach simulates the deformation of isochrones using a Lagrangian tracer method. This method uses results from previous ice-sheet model simulations (Sutter 2019) (3D velocity and ice thickness) and therefore by-passes the need to directly incorporate the tracer into the ice-sheet model. Direct comparisons are then made

between the elevation of the modelled and observed isochrones with a number of continuous layers dated between 38 ka and 160 ka.

Multiple processes combine to produce the observed isochrone stratigraphy (surface accumulation, basal melting, ice flow). The authors navigate this complexity by identifying regions of the ice sheet where individual processes dominant the isochrone deformation.

The authors conclude that in areas of slow ice flow far from the coast, such as ice divides, paleo surface accumulation can be determined using this method. Furthermore, isochrones are more accurately reproduced when the ice-sheet model is evolved transiently over the past ∼200 ka rather than maintaining the present-day ice flow and climate configuration. This highlights the importance of correctly simulating the past evolution of the ice sheet for model initialization.

In areas of faster horizontal ice flow, internal vertical flow is overestimated suggesting basal drag or topography is incorrectly characterized in the ice sheet model, possibly a result of the coarse resolution of the ice-sheet model. There are also more substantial difference in stratigraphy in areas where there is more complex historical variations in accumulation, i.e. change in synoptic systems rather than a simple temperature-accumulation relationship.

The authors argue that approaches similar to this, which compare modelled and observed internal stratigraphy, signal a step-change in paleo-ice-sheet modelling and future ice-sheet projections. They suggest a model intercomparison using this data is a possible future direction.

Overview: Although this is not the first time ice-flow models have been used to interpret ice-sheet stratigraphy, most previous studies have be restricted to ice-flow models covering relatively small spatial scales and analyzing a small number of radar profiles. Here simulations covering the whole Antarctic Ice Sheet are used to assess internal stratigraphy across an array of radar profiles covering a variety of locations with differing ice-flow, accumulation and basal conditions. A further novelty of this work is the use of a Lagrangian tracer applied to results from previous ice-sheet simulations. Not only is this efficient, but also has the potential to compare different ice-sheet models used for paleo ice-sheet simulations.

This work is presented as a starting point for more in-depth assessments in the future, and provides an interesting first step which will be of interest to a range of readers within the glaciology community (ice-sheet modellers, radar geophysicists and paleo glaciologists).

I found the work interesting and on the whole well written. I have included below some more substantial points that I believe should be addressed fully before final publication. I also include a commented PDF detailing line-by-line notes.

Main Comments: • I think the authors could be more quantitative with their assessment of how well modelled isochrones match observations. This is mentioned for some cases towards the end of the manuscript, but I think it should be more prominent throughout. After all, one criticism of traditional radar analysis may be it's qualitative nature. Here is a great opportunity to perform a more quantitative assessment, especially given the potential to apply this method further in the future.

• Lines 50-52: "Discrepancies in the initial state with respect to the actual real world ice-sheet can propagate and multiply during the model simulation due to the intrinsic nonlinearities of the system." Some attempts have been made to combat this using transient inversions, see Goldberg, D. N., Heimbach, P., Joughin, I., & Smith, B. (2015). Committed retreat of Smith, Pope, and Kohler Glaciers over the next 30 years inferred by transient model calibration. Cryosphere, 9(6), 2429–2446. https://doi.org/10.5194/tc-9-2429-2015

• Lines 72-73: "englacial isochrones have not been getting the required attention in the context of tuning targets for continental ISMs" - this is a very important point, which I feel should be more prominent in the manuscript. Maybe include some mention of

what these layers are traditionally used for, but their full potential is not being utilized.

• Isochrone stratigraphy is a result of the cumulative effects of surface accumulation, basal melting and ice flow. It would be good to have some comment on how these processes effect the stratigraphy generally and how they can be picked out from the structure of layers.

• Line 95: isochrones covering "38 ka to 170 ka" – is it not possible to trace earlier layers? I'm curious about this choice, wouldn't earlier layers have interesting histories too?

• Section 2.1: I found the inclusion of a detailed section summarizing the formation of isochrones to be well written and a nice addition to the text. It opens up the remainder of the manuscript to those who may be unfamiliar with ice-penetrating radar surveys.

• Section 2.2: • o The results of this work rely heavily on previous ice-sheet simulations from Sutter et al., 2019. It would be really helpful to have more details about these simulations. One particular question is; is GIA included in the model? You mention transient bedrock topography. o o It would also be helpful to have a summary of the differences between the model results in the three different cases used for the Lagrangian tracing; pal, pal-pd and pd. In particular what are the difference in pal-pd and pd with respect to the 3D velocity and ice geometry? Are there any clear differences which result in the different isochrone elevation. With respect to pal, to what extent has the ice flow/thickness varied? o • Fix typo in equation (1)

• Lines 178-180: "Misfits of the ice-sheet model state in terms of elevation and velocity field relative to the true (unknown) ice sheet state at that point in time in the past, will lead to deviations of the modeled isochrone as observed in the ice sheet today." Deviations between the modelled and the observed isochrones are a result of cumulative differences between the model and reality, not at a single point in time.

• Lines 191-193: "From this elevation map we then extracted the computed tracer-

, bedrock- and surface-elevation as well as the melting at the base of the ice and the corresponding geothermal heat flux (which was provided as input data) along the individual radar transects." This statement is a bit confusing. The elevation of the tracer is extracted from the tracing process, but all other parameters are taken from the ice-sheet model results.

 c Lines 196-197: "(i) climate forcing, (ii) model parameterisation, (iii) bedrock and geothermal heat flux." I agree with the assessment that these three processes affect ice-sheet internal structure and like the way you have gone about targeting them individually!

 c Lines 253-254: "The modelled isochrone elevations discussed above were computed on the basis of transient snapshots of local velocity and topography fields and show a good match to observed isochrone elevations." More details are needed here. What is the initial state? How much does this vary from present day? How does velocity vary in time? I assume very little? I realize these details are probably given in Sutter 2019, but it would be good to give a brief summary here. Especially given the next section of text.

 c Use of pd or pd-pd – which one do you want to use?

 c Lines 273-276: This is a really interesting point. Details of calibration may vary depending on desired analysis/end product. Most models used for near term projections (100-500 year) tend to use method (a). But clearly for projections over a longer period, and paleo simulations, the ability of the ISM to reproduce paleo-proxies is vitally important.

 c Line 295: "ice-sheet model parameterisations" I don't feel this point is sufficient explored in the text. What parameterisations are you referring to? Have you used models with difference ice rheology parameters, etc?

 c Lines 303-304: "dominating or confounding effect" - I agree that the combination

of these processes can make deciphering the true history complex. Do you have any insight into how to pick apart these contributing components?

â𝑋ć Figures 7 and 8 – these figures are under utilized and they are very few references to them in the text. I suggest you pick out some more details that the reader may find interesting and include them in the main text.

â𝑋ć Line 348: "Basal melting at the bed of the ice along the radar tracks is unfortunately unknown" Is there any evidence of water in radar profiles, i.e. flat bright reflectors, or isochrones that are drawn down and intersect the bed, that may suggest melting?

â𝑋ć Line 359: "the modelled isochrone" – can additional information be gather from the relationship between isochrones (modelled and observed) and also from the relationship between their misfits? Suggesting possible changes in time not included in the model.

â𝑋ć Lines 373-374: "synoptic activity can dominate the spatial and temporal variability in precipitation" I don't know too much about spatial and temporal patterns in accumulation, but is there significant inter-annual variability in regional climate models in these regions that could be used to give some additional bounds on the changes that could be expected?

â𝑋ć Line 380: "identify past accumulation patterns" How reliant is this on having additional data from ice-cores or good climate models?

â𝑋ć Lines 401-403: "While analysing the match of an ISM simulation with the internal stratigraphy is not as straight forward as using surface observables, it could improve both paleo ice-sheet reconstructions as well as sea level projections due to more realistic initial ice-sheet configurations." It would be good to detail what impacts (if any) an incorrect internal ice-sheet stratigraphy would have on future projections.

Figures: Most figures and captions need some attention to improve their readability. At present they have the potential to be really good, but need a little more work. I have

included comments in the attached PDF.

Individual Comments: I include an attached PDF with minor line-by-line comments. One reoccurring issue is the use of compound adjectives: ice-sheet model, present-day accumulation, etc. This should be addressed consistently throughout the text.

Please also note the supplement to this comment:
https://tc.copernicus.org/preprints/tc-2020-349/tc-2020-349-RC2-supplement.pdf

**Supplement:**

[revised manuscript text omitted]

---

## Author Comment (AC1) · 19 Mar 2021

We thank the reviewer for the very positive, detailed and constructive assessment of our manuscript. In the following we provide a point-by-point response (reviewer comments in black, replies in blue, changes to the text in non-serif). At the end of the document you find the track-changes manuscript (new text in blue, modified text in red).

Main Comments:

1. The title
The title should be tightened to be more specific. E.g. "Simulating the internal structure" could be qualified, especially given that the isochrones were derived from simulated velocity and geometry fields, rather than calculated online in the ice sheet model. Suggest something like: "Investigating the internal structure of the Antarctic Ice Sheet: the utility of isochrones for spatio-temporal ice sheet model calibration" OR "On the use of isochrones as a novel diagnostic for ice sheet model performance"

We thank the reviewer for the nice suggestions and agree that the title should be more specific. We adopted suggestion number 1 and the title now reads:

"Investigating the internal structure of the Antarctic Ice Sheet: the utility of isochrones for spatio-temporal ice sheet model calibration"

2. Structure and readability of sections 3-5:

I found the structure and flow of sections 3-5 difficult to follow, and because of this the findings of the study – particularly with respect to clearly identifying the sources of mismatches between observed and simulated isochrone depths – and their significance were weakened. For example, L195-206 introduces the main goal and three main potential sources of model-obs mismatches. While the sections that follow contain discussion of each of these sources, it is not always clear which source is being considered (e.g. with subsection headers), and there is some repetition (particularly sections 3.2 and 4). It'd be good to see a restructuring through sections 3 and 4, with distinct subsections to systematically address each source of mismatch, first with respect to Dome C (section 3), then with respect to the broader EAIS (section 4).
In each section, the aim should be named early in the paragraph so that it's clear why certain experiments are being assessed, and the significance of the results. E.g. for L253-276, evaluating the model parameterisation, the aim of that paragraph is stated on L262-263, but should be moved to the start. This would help in interpreting the results.
In section 4 on P16 L341-376, the presentation of the results/physical conditions is mixed up with the discussion of the processes that could contribute to obsmodel mismatches and how they contrast regionally. It'd be good to first clearly discuss the Dome C and Dome Fuji results. E.g. "for DML-VIII23, the bedrock elevation is above sea level and relatively flat. For this transect, the obs-model mismatch..." Then in a new paragraph, contrast the obs-model mismatch between each of the transects (DML-VIII23, CEA-10, DML-IV24) to highlight the sources/processes that could contribute to the mismatch. It'll then be easier to come away with a clear picture of what is causing what on the larger scale.

It'd be helpful for readability of the conclusions section to restructure into 4 paragraphs:

• Summarise how well the paleo model outputs match the main observed features of the internal ice sheet structure and capture broad-scale SMB

patterns.
• Summarise how the method helps to understand the processes underlying/
sources of mismatches between observed and simulated isochrone
depths, with specific reference to the four areas identified in comment 1.
above: (a) boundary conditions; (b) forcing time series; (c) initial conditions/
parameters; and (d) simulation time.
• Summarise where more efforts are needed: (a) constraints on the spatial
variability in paleo accumulation rates; (b) constraints on basal drag in marine
sectors.

• Discuss the model intercomparison

We agree with the reviewer that sections 3-5 where difficult to follow and did a complete overhaul.
Section 3 is now divided into four subsections titled:

3.1 Dome C - evaluating the paleoclimate forcing (i)

3.2 Dome C - impact of paleo spin-up and model parameterisation on simulated isochrone
elevation (ii)

3.3 Dome C - Impact of lower boundary conditions on isochrone simulated isochrone
elevation (iii)

3.4 Caveats to modelling isochrones with large scale ice-sheet models

The figures in section 3 were modified according to both reviewers' suggestions and Figure 6 (now
Figure 7) complemented with the results of a present-day parameter study illustrating the effect of
different parameterizations affecting sliding on isochrone elevation. The present-day ensemble is
described in section 2 of the ms.

Figure 8 (still Figure 8) was modified and now consists of a panel illustrating the effect of different
geothermal heat flux choices on the simulated isochrone elevation as well as a panel showing the
along-transect geothermal heat flux and computed basal melt.

Section 4 is now divided into two parts

4.1 Simulated isochrone elevation along the Talos Dome - Lake Vostok - transect.

4.2 Simulated isochrone elevation along the Dome Fuji - Dronning Maud Land - transect.

We added an additional Figure showing the simulated isochrone elevations for a present-day
parameter study akin to Figure 7 in section 3 but for transect CEA-10 (90 ka isochrone)

Conclusions

We followed the reviewer's suggestion to compartmentalize the discussions which is now arranged
into four parts covering the different impacts on the simulated isochrone elevations:

-climate forcing

-ice sheet model parameterization

-effect of paleo-spin up

-geothermal heat flux and bedrock elevation

The fifth item shortly introduces the planned model intercomparison.

3. Reporting of basal drag and methodology

One of the main findings is that the basal drag can play a large role in the mismatch
between observed and simulated isochrones. I.e., the large mismatch in
some regions is due to an overestimation of vertical velocities where there is low
basal friction. However, the actual basal drag is not reported in this paper for
the simulations. This makes it very difficult to ascertain whether the aims of the
study that relate to basal drag have been adequately addressed, and the degree
to which the discussion around the accuracy and constraints on the basal drag
applies. The basal drag coefficient should be reported for each of the simulations
as a separate figure in the main body of the manuscript. It'd also be good to see
the drag coefficient as a separate panel on figure 9. I recommend adding supplementary
material that describes how the basal drag was calculated for each
of the simulations.
It'd also be worth reiterating some of the salient features of the model
setup/methodology, particularly those aspects that are relied upon for interpretation
of the results. For example, how was the model tuned at Dome C? This is
key in understanding point (i): the impact of the paleoclimate climate forcing on the model-obs
mismatch (see specific point below).

We agree with the reviewer, that a more detailed description of the model features and settings is
necessary. We therefore expanded section 2.2 considerably and hope this will provide the reader with
a better perspective on how the model estimates sliding. We decided not to include figures of the
spatial distribution of the yield stress in the paper because we do not know the yield stress in reality.
Thus, a direct comparison between model and observations is not possible. Moreover, it is not the
yield stress at present day which is relevant for the elevation of the observed isochrones but its
integrated effect throughout the time since deposition of the isochrone. This means that a present-day
comparison does not capture this.

4. Generalisation of results to the large-scale

I'm slightly concerned that the abstract (and then the conclusions) oversells the
results by use of some general terms e.g. "Antarctic Ice Sheet", "the interior", and
"subglacial basins". The results for the isochrones that are analysed certainly support the
interpretations; however, in reality, these isochrones only cover very small parts of the East
Antarctic Ice Sheet. How representative are the results for these particular observational
isochrones for the large-scale ice sheet?

The reviewer brings up an important aspect of using englacial layers to benchmark paleo-ice-sheet-
models: the availability and spatial extent of dated isochrones. In contrast to Greenland, dated
isochrones in Antarctica are sparsely distributed and predominantly cover regions around deep ice
cores, ice divides and domes. While this is certainly a good start, at the current stage it is not possible
to carry out a model-based pan-Antarctic isochrone study as there are not enough picked and dated
radar transects available. While we analysed one radar product for West Antarctica (provided by
Ashmore et al.) we did not include the results in this paper, as the data is limited to the Holocene and
our paleo-ice-sheet model simulations do not resolve the 0-10 ka BP grounding line position in this

sector of the ice sheet sufficiently well to carry out a meaningful comparison between observations and model results. We also looked at a recently published set of dated isochrones from Titan Dome (Beem et al. 2020 TC). For this particular geographic setting, the bedrock topography of the Bedmap2 data at 16 km which we use as an input data set for the ice sheet model is drastically different from the more recent radar observations at high resolutions (see Figures below in an example for the 16.8 ka isochrone and another transect for the 59.4 ka isochrone). We would prefer to keep the WAIS radar transects and the Titan Dome case out of the manuscript as it is very lengthy already. The Titan Dome example is interesting but does not necessarily add new insights as the impact of bedrock uncertainties is already discussed for Dome C. We hope to include the data from Ashmore and Beem in the intended model intercomparison.

[Figure]

Figure 1: Observed and modelled isochrone along a transect at Titan Dome (observed isochrone and high-res bedrock topography from Beem et al. 2020). The simulated isochrone is depicted in red, the observed isochrone in black.

We changed the wording in the abstract from:

"We show, that paleoclimate forcing schemes commonly used to drive ice-sheet models work well in the interior of the Antarctic Ice Sheet and especially along ice divides, but fail towards the ice-sheet margin."

To

"We show, that paleoclimate forcing schemes derived from ice core records and climate models commonly used to drive ice-sheet models work well in the interior of the East Antarctic Ice Sheet and especially along ice divides, but fail towards the ice-sheet margin in the studied cases"

these isochrones only cover very small parts of the East Antarctic Ice Sheet.

We would like to note that the isochrones analysed in this manuscript span a good portion of the East Antarctic Ice Sheet and make use of the available pool of dated isochrones. The region we cover extends from Dronning-Maud-Land to Dome Fuji, from Lake Vostok over Dome C towards Talos Dome. We simulate the isochronal planes for all regions of the Antarctic Ice Sheet with an ice thickness above 2200 m and a velocity smaller than 7 m/a (see Figure 2 and 4). We could have plotted some "artificial transects" along these planes to expand the coverage discussed in the paper, but there are unfortunately no observations to compare these simulations to. We do hope that efforts such as AntArchitecture and our paper will motivate future radar missions into hitherto unexplored regions of Antarctica. We added a sentence in chapter 3, pointing out that:

"We would like to note, that while we simulate isochronal layers throughout East and West Antarctica our model-observation-intercomparison is mostly limited to ice domes and along ice divides in East Antarctica as only these regions are covered by dated radar-based isochrone observations. Future radar explorations in Antarctica will hopefully complement the available data by observations away from ice divides and along drainage sectors as these are the regions which would point to critical misrepresentations in ice sheet model simulations."

5. Grammar and spelling:
I have noted some grammatical and spelling points below, but not all of them –
others should be able to be corrected fairly easily with a standard LaTeX spelling
checker or online e.g. grammarly.com. Some of the sentences are also long and
a little bit wordy – it would be worth shortening the longer sentences for improved readability.

We are sorry for grammatical and spelling issues and thank the reviewer for pointing them out. We hope we caught all mistakes.

Specific Comments

P1 L6. "We simulate observed isochrone elevations within the AIS via passive Lagrangian tracers" >> "We calculate isochrone elevations from simulated AIS geometries and velocities via passive Lagrangian tracers".

Done.

P2 L31. Check citation style. Also on P6 L152-153, P12 L283, P18 L367.

Done.

P2 L45. What does "in scope of" mean here? "Relevant to"?

Yes, edited accordingly.

Table 1. I couldn't see the analysis of a number of these isochrones (e.g. CEA-7,8,12,13). Were these reported within the figures and text?

Thanks for pointing this out, the caption of the table now reads:

Overview of analysed isochrones. Transects presented in this work are DC - X45a, X57a,Y77a,Y90a, CEA-10, DML VIII-23/IV-24

P6 L134. Sentences starting from "We use" to a new paragraph.

Done

P6 L136-139. The sentence starting with "Based on the analysis" was a bit confusing. I took it to mean the following: (a) that the observed isochrone data used in this study (derived from Winter et al. 2017) are assumed to have a maximum age uncertainty of 1 ka; (b) that the observed isochrone data are all above or at 2000 m depth; (c) that the age uncertainty nonlinearly increases with depth; (d) that the age uncertainty of the observed isochrone data is always lower than the uncertainty in the simulated data. Is this what is meant? This sentence could be reworded for improved clarity.

"We use various internal layers in this study. Details on the radar resolution, dating methods and resulting uncertainty can be found in the respective original publications (Cavitte et al., 2016;Winter et al., 2019; Leysinger Vieli et al., 2011). Based on the analysis performed on five different radar systems by Winter et al. (2017) near EDC, we consider a maximum age uncertainty of around 1 ka for each isochrone above 2000 m depth. (roughly 2/3 of the ice thickness) (Winter et al., 2019). This range covers most of the horizons considered in this study, so we can transfer the uncertainty of 1 ka. Below a depth of 2000 m, the age uncertainty increases non-linearly with depth towards the bed (Winer et al., 2019). At places where the deepest ice is younger than the ice around Dome C, the age gradient with depth will then be less steep towards the bed than the one determined at Dome C. Thus, the age uncertainties of the horizons below 2000 m depth will be lower than at Dome C. We consider this uncertainty always to be smaller than the proposed age derived from the ISM data."

P6 L138. "2/3 or" >> "2/3 of"

Done

P6 L154. "relatively coarse resolution" >> "relatively coarse resolution model grid"?
Does the mesh size evolve over time with grounding line migration or is it static? Some more details on the model experiments would be helpful here.

Yes, we refer to the model grid here. We edited the text and clarify that we use a static model grid:

"We employ a combination of the shallow ice (SIA) and shallow shelf (SSA) approximation (SSA+SIA Hybrid Winkelmann et al., 2011a) with a sub-grid grounding line parameterisation (Gladstone et al., 2010; Feldmann et al., 2014) to allow for reversible grounding line migration despite using a relatively coarse resolution static model grid of 16 km (Feldmann et al., 2014)"

P8 L175. How is "accurate enough" determined? How do we know that the misfits between the radar isochrones and the simulated isochrones using this method and the ISM are not sensitive to the temporal and spatial resolution of the ISM output and the Lagrangian particle tracking algorithm? It would be good to see a sensitivity analysis or uncertainty quantification here.

We thank the reviewer for this suggestion. The manuscript now contains a figure showing the effect of different temporal resolutions of the ISM output and different total tracer numbers on the isochrone elevation (new figure 3 on page 10 and corresponding text on page 9).

"Velocity snapshots between 1 ka and 10 ka largely produce the same isochrone elevation (see Figure 3)."

"Figure 3 illustrates the changes in the computed isochrone elevation due to different numbers of tracers. In regions where homogenous flow dominates, the number of tracers does not affect the simulated isochrone elevation much. However, if spatial gradients in ice-flow and topography increase, simulated isochrone elevations diverge with low tracer densities."

As we do not have paleo simulations on other resolutions than 16 km, we could not test the impact of the ISM output resolution which can be significant in faster flowing regions due to the overall dependency of ice sheet model simulations on resolution. However, this effect is probably relatively small in regions such as Dome C.

P8 L184 "We also tested other seeding strategies…" What was the outcome of these tests (i.e. was there any sensitivity to seeding mask)? A section quantifying the uncertainty in the tracer method would be appropriate in a supplementary material document

We would like to forego an additional supplementary material covering tracer seeding considerations as this would be rather technical and does not add any insights with regard to the main findings of the manuscript. We added a paragraph shortly motivating our choice of seeding mask.

"The choice of seeding mask is solely motivated by the tracer coverage. As long as the transects of interest are covered by the seeding mask, the latter only affects the density of tracers (a larger seeding masks leads to sparser tracer coverage if the number of tracers is not adjusted accordingly). We found that a seeding mask defined by the ice thickness and surface flow considerations mentioned above provides the best coverage given the radar transects discussed in this study."

P8 L201. "the model ensemble was tuned". What does this mean? Please provide specific details of what fields were tuned and to what data.

The methods section was expanded by a more detailed description of the model setup (including a discussion of the relevant tuning parameters) also including a sentence regarding the considerations behind the model tuning (see also response to comment on P9L214):

"[…] Both the paleo simulations and the present day equilibrium simulations where tuned to match the observed present day surface elevation (with a focus on the deep ice core sites), ice volume and grounding line position."

[…] "Here, we will primarily focus on Dome C as ice sheet model parameters relevant for ice flow and basal friction where tuned to match the regional ice-sheet configuration."

„Table 2 provides an overview of the friction and sliding parameters chosen for the 2 Ma, 220 ka and present-day model ensemble. In PISM the till friction angle φ controls the yield stress at the ice-bedrock interface which can be set to be a function of the bedrock elevation (increasing with elevation). The yield stress is determined by

(see ms for correctly set eqn.)

where φ is the bedrock elevation dependent till friction angle and $N_{till}$ the effective pressure. As in Sutter et al. (2019) we choose a pseudo-plastic power law with the parameter q controlling the amount of sliding via the relationship

(see ms for correctly set eqn.)

where $\tau_b$ is the shear stress, u ice velocity and $u_{threshold}$ the threshold velocity at which $\tau_b$ has the exact magnitude as $\tau_c$ (condition for sliding).The so called SIA enhancement factor in the 220 ka and present-day ensemble is 1.0 as in Sutter et al. (2019)."

"For further details regarding the ISM setup see section 2.1 in Sutter et al. (2019). Geothermal heat flux is taken from Shapiro and Ritzwoller (2004) as well as from Purucker (2013); An et al. (2015) and Martos et al. (2017). Topography data are from Bedmap2 (Fretwell et al., 2013) except in the present-day simulation where we use the new BedMachine Antarctica dataset from Morlighem et al. (2020). We model isochrone elevations (see 2.3) on the basis of full paleo-ice-sheet model runs (pal, model integration from 2 Ma (Sutter et al., 2019) and 220 ka BP in this study), the present-day snapshot of the latter (pd-pal) and a present-day equilibrium ensemble (pd, with an integration time of 2000 years following a thermal spinup in a fixed ice sheet geometry for 200 ka). The 2 Ma simulations in (Sutter et al., 2019) are initialised at 2 Ma BP from a present-day ice sheet geometry. The 220 ka simulations where initialised from the 220 ka BP output of the 2 Ma simulation with isochrones evolving until present day according to the computed transient ice flow starting at their respective age (see Table 1) in the past. In the present-day snapshot (pd-pal) and present-day equilibrium ensemble (pd) isochrones evolve on the basis of the simulated present-day flow."

P8 L201-202. "To assess the impact of model" >> "To assess the impact of (ii) model"

Done.

Figure 3. For this figure and for figures 4, 6, 8, and 9, please make the figure bigger (e.g. textwidth) and increase the font size. The blue lines in panel C are difficult to see - perhaps use black instead - and it'd be great to label the transects in panels B and C (see also comment on figure 4).

The colouring of the lines in C correspond to the observed normalised elevation above bedrock along the transect. We modified the figure caption to make this clearer. There are no transects in panel B as there is no 38 ka isochrone for this region. Transects in C are now labelled.

In the caption, I wasn't sure what this sentence meant: "Due to the small mismatch… are discernible." Does this mean that the background colour is relatively uniform? Consider modifying the colour bar to zoom to the relevant colour range represented in the figure. Also, "strong" >> "large" in the second last sentence of the caption.

We agree that this description is vague. We reformulate this part of the caption which now reads:

"The mismatch between the simulated isochronal layer and the observed isochrone elevation along the transects is generally small in the Dome C region with the exception of the upper right section in panel C) where the simulated isochrone is off by up to 40% of the local ice thickness."

P9 L214. "To evaluate the validity of the forcing approach in Sutter et al. (2019).."
Earlier, it is mentioned that the model ensemble is tuned to match the regional configuration near Dome C. For what paleoclimate forcing was this tuning carried out? Please comment on whether/how this tuning might impact the capacity to assess the validity of the forcing approach.

For the 220k runs, the same paleo-forcing as in Sutter et al. (2019) was used but with different precipitation-temperature scaling. We modified the text accordingly:

"It is important to note, that the model parameters chosen in Sutter et al. (2019) where tuned using a climate forcing with a temperature-precipitation relationship of 3%. The paleo-simulations created for this paper employ identical model parameters and the same forcing but with temperature-precipitation relationships of 5, 6 and 8 %$K^{-1}$), see section 2.2."

Figure 4. The transect lines and labels are difficult to see on panel A - please make the lines thicker and more contrasting with the background. It's difficult to determine which line is which on panel B (the brown/purple/red colours are similar - perhaps choose more contrasting colours for the bed elevation). Please also make the lines thicker and include a legend on panel B?

Done.

P10 L219. "DC-57a" is not marked in bold in figure 4A (that's DC-X45a). Which transect is referred to here? Suggest labelling all transects more clearly on the figures.

Corrected.

P10 L221. Why use 3, 5, 6, and 8

This is motivated by estimates of the Dome C paleo temperature-precipitation relationship (Frieler et al. 2015) as mentioned in the text. The sentence now reads:

"Isochrones are computed from the output of experiment B1- P1 in Sutter et al. (2019) using a scaling constant between temperature and accumulation anomalies (i.e. percent change of accumulation for every degree of surface air temperature change) of %3$K^{-1}$, as well as new simulations with scaling constants of 5, 6 and 8 %$K^{-1}$, inspired by the approximate range indicated by paleo proxies of 5.9±2.2%$K^{-1}$ (Frieler et al., 2015)."

P10 L225. "completely reproduces". What does this mean? The red line in panel B is sometimes outside of the 5-6

We agree that this is misleading. We reworded the sentence to:

"According to our simulations a precipitation scaling between 5 and 6 %K$^{-1}$ reproduces the 96 ka isochrone best, which is in accordance with an ice core based relationship of 5.9±2.2%K$^{-1}$ for EDC (Frieler et al., 2015). "

P10 L227. "at least for the interior of the East Antarctic Ice Sheet". The results show that is valid for Dome C (and given that the model is tuned for this region). It's not clear that this conclusion can be extended to the interior of the EAIS in general. Surely this depends on the degree to which the climate forcing is appropriate in other regions? (which is indeed addressed in section 4).

We agree! This conclusion cannot be simply extrapolated to the whole interior EAIS. We moderated the sentence accordingly:

"This gives us confidence that the paleo mass balance forcing approach in Sutter et al. (2019) is valid at least for the region around Dome C and likely for the larger parts of the interior East Antarctic Ice Sheet (see section 4) "

P10 L230-231. This result might be related to the fact that the method employed in Martos et al. (2017) is not physically realistic. For a description and an updated GHF product for all of Antarctica, see: Stål, T, et al. "Antarctic geothermal heat flow model: Aq1." Geochemistry, Geophysics, Geosystems: e2020GC009428, https://doi.org/10.1029/2020GC009428.

Thanks for pointing us to this study. We would have loved to compare the four geothermal heat flux compilations used in our simulations with the one from Stål, however the Pangaea link they provide in their paper is dead (DOI not found) and after a manual search in Pangaea for the data it turned out that the data is not yet available (dataset in review, accessed 09.03.2021 https://doi.pangaea.de/10.1594/PANGAEA.924857). We will use this new data set in future studies.

P11 L242-252. I found this paragraph difficult to follow. First, the discussion of the 0.5 cm/a difference between observations (Stenni et al., 2016) and the models simulations + RACMO could be clearer. Is there a reason the simulations match RACMO but not obs? Second, for clarity please reference panel B in the text (e.g. "When we compare (figure 5B) our simplified: : :") and describe the bias correction that is used. Consider restructuring the paragraph for clarity, and add labels for the different curves in figure 5A.

We agree that this paragraph is not very clear. We hope the revised paragraph is easier to follow. The climate forcing used in the simulations is created from climate time slice anomalies from the Last Interglacial, Last Glacial Maximum and Pre-Industrial. In between the time slices, we interpolated the climate linearly based on the variations in the Dome C deuterium record. The resulting transient anomalies were added to RACMO which is used as the present-day reference forcing. This is the reason why the simulations match RACMO but not obs. For present day.

P11 L251. "along ice divides" >> "along the ice divide near EPICA Dome C"

Done.

Figure 5. What are the semi transparent solid lines in panel A? Bias corrected values

from the simulation? Please describe, along with the dashed lines, in the figure 5
caption.

Correct, these are the bias corrected values from the simulations. We decided to remove them as they distract from the actual forcing used in the simulations.

P12 L253. "where" » "were"

Thanks, done throughout the ms.

P12 L256. Here 5

P12 L262-276. Should "pal", "pd-pal", and "pd-pd" be in italics here and through this
paragraph?

We changed the font from italics to regular throughout the paragraph.

P12 L267-276. This is a really neat and interesting result. It's also interesting because
presumably unknown parameters (e.g. the basal friction coefficient) are also somewhat uncertain in
the initialisation of

We are happy, that the reviewer likes the result.

P12 L278-279. "are small" >> "are generally small". E.g. the difference between obs
and sim isochrones for 38 ka along DC-Y77 between _15-25 km is much larger than
for any other isochrone in this portion of the transect.

Done

P12 L286. "antarctic" >> "Antarctic". Here and elsewhere.

Done

P13 L293. "60 and 90 ka" >> "90 and 60 ka". How much uncertainty does the shorter
spin-up time for 90 and 60 ka introduce?

This is a very good question and I'm afraid we cannot say for sure. In fact, we think the impact might be small as the 220 ka simulations where initialised from the 2Ma simulations in Sutter et al. 2019, the only difference being the temperature-precip scaling. To address this, we would have to rerun the experiments for the 130 ka isochrone with an experiment initialised 30 ka earlier. Due to time restrictions and limited computational resources, we would rather not do this at this stage. But it is certainly something to keep in mind for future experiments. We decided to removed the sentence:

"Finally, the simulations analysed in the previous sections where initialised at 220 ka BP. Therefore the model spinup for the calculations of the 130 and 160 ka isochrone elevations was only 60 and 90 ka, respectively. This could be another aspect influencing the particularly poor match between modelled and observed isochrone elevations for these ages."

As the word "spinup" might be misleading here and we cannot quantify the uncertainty introduced.

P13 L297-298. A bit of a jumbled sentence. Suggest: "Areas with sparse radar observations
may have bed elevation estimates that differ from high-res radar data by
several hundred metres."

Thanks, and Done.

FIgure 7. Final line: "The coloured bars on the bottom of" >> "The coloured bars at the bottom"

Done.

P14 L310. "focus of an upcoming paleo-ice-sheet model intercomparison". Great idea.

Thanks ☺

Figure 8. In the second last line of the caption, "normalised with" >> "normalised by"

Done.

P15 L322. "The northern half of the transect bottom part in Figure 9". It's not hugely clear which segment of the transect this refers to - perhaps add demarcation on panels A and B of figure 9.

We agree, done and rephrased.

"The northern half of the transect close to Talos Dome (bottom left in Figure 10) is dominated by the imprint of the Wilkes Subglacial Basin with a large misfit between the modelled and observed isochrone (≈ 26% RMSE along km 0-500 in Figure 10, compared to ≈ 8.3% and ≈ 4.8% along km 500-1000 and 1000-15000, respectively). "

P16 L334. "The basal friction in the model is a function of bedrock elevation". Please provide the equation and description of the basal friction calculation, perhaps in supplementary material.

We now provide the formulation of the yield stress and its relation to the relevant parameters for sliding in section 2.2

P16 L347. "isorchrone" >> "isochrone"

Thanks for spotting this, corrected.

P16 L349. "We limit ourselves to…" New paragraph.

Done.

P16 L356-358. Move "which encompasses...Wilkes Subglacial Basin" to earlier where CEA-10 results are first introduced. Reword remaining part of sentence: "The comparison between DML-VIII23 and CEA-10 potentially highlights a methodological deficiency (leading to unrealistic internal flow and basal sliding) as the isochrone mismatch in CEA-10 cannot be remedied by a surface mass balance correction."

Done.

P16 L365. "Dome C". Should this be Dome F or Dome C here and also on page 18?
I'm not sure I understand this argument if it's Dome C.

We agree that the reference to Dome C is confusing here. What we meant to say, is that the temperature-precipitation relationship at EDML might strongly differ from the continental mean (ca. 5%/K). We rephrased both sentences:

"One potential reason could be a relationship between temperature and precipitation anomalies in the EDML region which strongly differs from the continental scale temperature precipitation scaling of ca. $5\pm1\%K^{-1}$. This would affect the surface mass balance forcing and therefore the elevation of the isochrone. However, the proxy-based paleo-precipitation-temperature relationship at EDML is very similar to the continental mean albeit with considerable uncertainties of ±2.8% for EDML (Frieler et al., 2015)."

Figure 9. Consider a vertical line on panel B indicating the Dome C location so that we can compare GHF here to the northern portion of the transect. Consider also demarcating (e.g. with a coloured vertical line) the "northern portion/bottom half" of transect in panel B, and perhaps with two "x" the same region on panel A. For panel C, here and in figure 8, I suggest using a different colour map for the basal melt rate - the blue-white-red transition is usually used for differences (where white is 0). In the second line of the caption: "beige" looks more like dark red to me.

Done.

P18 L379-380. It would be better to reword as follows: "We are able to reconstruct most large-scale englacial layer features of the observed isochrones..."

Done.

P18 L394. "A model intercomparison". New paragraph

Done.

Figure 10. Should "DMLVIII 22" on panel A be "DMLVIII 23"? It's hard to see the dashed lines on panel B – please make the lines thicker.

There's transect DMLVIII-22 and DMLVIII-23, both are marked correctly. We made the Bedmachine lines in panel B thicker.

P20 L405. "This would facilitate" >> "This would facilitate the evaluation of"

Done.

[revised manuscript text omitted]

---

## Author Comment (AC2) · 19 Mar 2021

We thank the reviewer for the very positive, detailed and constructive assessment of our manuscript. In the following we provide a point-by-point response (reviewer comments in black, replies in blue, changes to the text in non-serif). At the end of the document you find the track-changes manuscript (new text in blue, modified text in red).

Individual Comments: I include an attached PDF with minor line-by-line comments.

Thank you very much for the detailed annotations in the attached PDF! Please find changes to the manuscript as well as new figure 3 and 11 in the track-changes ms at the end of this document (red and blue text).

Main Comments:

I think the authors could be more quantitative with their assessment of how well modelled isochrones match observations. This is mentioned for some cases towards the end of the manuscript, but I think it should be more prominent throughout. After all, one criticism of traditional radar analysis maybe it's qualitative nature. Here is a great opportunity to perform a more quantitative assessment, especially given the potential to apply this method further in the future.

We thank the reviewer for pointing this out and add root-mean-square-difference (RMSD)-numbers in section 3 (transect X45a), 4 (transect CEA-10), as well as for the modified figure 6 (now figure 7) which also consists of a newly introduced present day parameter ensemble. Please find a description of the pd-ensemble and a revised discussion of the ice sheet model and forcing in section 2.2 (see also responses to Reviewer 1.)

Section 3:

"According to our simulations a precipitation scaling between 5 and 6 %K$^{-1}$(RMSD of ≈ 3.8% and ≈ 3.3% for precipitation scaling of 5 and 6 %K$^{-1}$, respectively) reproduces the 96 ka isochrone best, which is in accordance with an ice core based relationship of 5.9±2.2%K$^{-1}$ for EDC (Frieler et al., 2015)."

Section 4:

"The northern half of the transect close to Talos Dome (bottom left in Figure 10) is dominated by the imprint of the Wilkes Subglacial Basin with a large misfit between the modelled and observed isochrone (≈ 26% RMSD along km 0-500 in Figure 10, compared to ≈ 8.3% and ≈ 4.8% along km 500-1000 and 1000-15000, respectively)."

"We show this by computing the 90-ka isochrone elevation in a paleo-simulation with a temperature-precipitation relationship of 8%K$^{-1}$ which cannot mitigate the drop in elevation in the first 300 km of the transect and exacerbates the deviation between simulated and observed isochrone elevation along km 500-1500 (≈ 16.7% RMSD along km 0-500 in Figure 10, compared to ≈ 8.7% and ≈ 7.8% along km 500-1000 and 1000-15000, respectively)."

We also introduce the metric based on which we analyse the differences in observed and modelled isochrone elevation at the end of section 2:

"Our main goal is the identification of systematic mismatches between predicted and observed isochrone geometry. As a metric for the difference between observed and modelled isochrones we use their respective elevations in the ice sheet above the ice-bed interface, normalised by the local ice thickness. This yields the root-mean square difference (RMSD) in %."

Discrepancies in the initial state with respect to the actual real world ice-sheet can propagate and multiply during the model simulation due to the intrinsic nonlinearities of the system." Some attempts have been made to combat this using transient inversions, see Goldberg, D. N., Heimbach, P., Joughin, I., & Smith, B. (2015). Committed retreat of Smith, Pope, and Kohler Glaciers over the next 30 years inferred by transient model calibration. Cryosphere, 9(6), 2429–2446. https://doi.org/10.5194/tc-9-2429-2015

Thanks for pointing us towards this interesting study, it is now referenced on page 2.

"To counter the effects of overfitting, promising attempts to improve the initialisation of ice sheet models have been made which involve transient inversions of multiple surface elevation observations over time (Goldberg et al., 2015) albeit only on the regional scale and limited to the extent of the satellite record."

"englacial isochrones have not been getting the required attention in the context of tuning targets for continental ISMs" - this is a very important point, which I feel should be more prominent in the manuscript. Maybe include some mention of what these layers are traditionally used for, but their full potential is not being utilized.

We agree, and this is meant to be a key message of the manuscript. However, we believe that the paragraph preceding the referenced sentence explores the model applications of isochrones, in previous years, sufficiently.

Isochrone stratigraphy is a result of the cumulative effects of surface accumulation, basal melting and ice flow. It would be good to have some comment on how these processes effect the stratigraphy generally and how they can be picked out from the structure of layers.

We added a sentence regarding the impact local basal melt in section 2.1 "Observation of Isochrones". The effect of surface accumulation and ice flow is also briefly introduced in this section.

isochrones covering "38 ka to 170 ka" – is it not possible to trace earlier layers? I'm curious about this choice, wouldn't earlier layers have interesting histories too?

Absolutely! The reasoning behind our choice of 38 ka – 170 ka was that we have a pool of dated isochrones available in the literature from various regions covering the same time scale and dated against consistent ice core chronologies. Isochrones older than 160 ka are available for the DC transects but not along the CEA and DML transects. We also analysed the recent compilation from Ashmore et al (2020) which covers a region straddling the Filchner-Ronne Ice Shelf. The oldest isochrone in this data set is 6.4 ka old. This data set is terrific to constrain the regional WAIS Holocene model-behaviour and we plan to fully utilise this in the planned model intercomparison and future work. However, the simulations we have available at the current stage do not resolve Holocene grounding line positions in this sector of the WAIS well enough to allow for a meaningful model data intercomparison. The second data set we analysed but not discussed in the paper is from Titan Dome (Beem et al. 2020). The spatial extent of the transects is similar to the ones we use at Dome C (Cavitte et al. 2020). However, the difference between the high-resolution radar bedrock reflector and the 16 km Bedmap2 data we use for our model is rather large (see figures below).

[Figure]

Figure 1: Observed and modelled isochrone along a transect at Titan Dome (observed isochrone and high-res bedrock topography from Beem et al. 2020). The simulated isochrone is depicted in red, the observed isochrone in black.

We amended the manuscript to mention these data sets as well (see also revised table 1) and explain our choice of the time span 38-170 ka.

"Unfortunately, the number of published and available traced and dated Antarctic isochrones is limited at the current stage, a situation which initiatives like AntArchitecture aspire to improve in the coming years. For this intercomparison we make use of three compilations of dated isochrones (Leysinger Vieli et al., 2011; Cavitte et al., 2016, 2020; Winter et al., 2019) which provide about 10,000 km of analysed radar transects with dated isochrones covering the past 38 ka to 170 ka BP (see Table 1). We focus on this time period as dated isochrones from this range are available for all regions considered here. Both younger and older dated isochrones are available e.g. for Dome C, Titan Dome (Beem et al., 2020) and for West Antarctica (Ashmore et al., 2020) (see Table 1)."

I found the inclusion of a detailed section summarizing the formation of isochrones to be well written and a nice addition to the text. It opens up the remainder of the manuscript to those who may be unfamiliar with ice-penetrating radar surveys.

Thank you very much :)

The results of this work rely heavily on previous ice-sheet simulations from Sutter et al., 2019. It would be really helpful to have more details about these simulations. One particular question is; is GIA included in the model? You mention transient bedrock topography. o o It would also be helpful to have a summary of the differences between the model results in the three different cases used for the Lagrangian tracing; pal, pal-pd and pd. In particular what are the difference in palpd and pd with respect to the 3D velocity and ice geometry? Are there any clear differences which result in the different isochrone elevation. With respect to pal, to what extent has the ice flow/thickness varied? o ã˘A ´c Fix typo in equation (1)

We agree! We expanded the discussion of the model simulations considerably and also explicitly state how bedrock elevation changes are modelled. There is no difference in the pal and pal-pd final ice sheet configuration, as pal-pd simply uses the last time slice of the pal simulation. The pd experiment was replaced with a pd parameter ensemble which is illustrated in figure 7 (transect DC-X57 a) and figure 11 (transect CEA-10). The ice sheet elevations of the individual simulations are plotted in figure 7 and figure 11. Thank you for spotting the typo in equation (1)!

[revised manuscript text omitted]

Misfits of the ice-sheet model state in terms of elevation and velocity field relative to the true (unknown) ice sheet state at that point in time in the past, will lead to deviations of the modeled isochrone as observed in the ice sheet today." Deviations between the modelled and the observed isochrones are a result of cumulative differences between the model and reality, not at a single point in time.

Absolutely right. The sentence now reads:

Misfits of the ice-sheet model state in terms of elevation and velocity field relative to the true (unknown) ice sheet state at that point in time in the past and throughout the paleo simulation, will lead to deviations of the modelled isochrone as observed in the ice sheet today.

From this elevation map we then extracted the computed tracer-, bedrock- and surface-elevation as well as the melting at the base of the ice and the corresponding geothermal heat flux (which was provided as input data) along the individual radar transects." This statement is a bit confusing. The elevation of the tracer is extracted from the tracing process, but all other parameters are taken from the ice-sheet model results.

We agree, the sentence now reads:

From this elevation map we then extracted the computed tracer-elevation. From the ice sheet model output we retrieved the bedrock- and surface-elevation, the melting at the base of the ice and the corresponding geothermal heat flux (the latter being derived from the input data) along the individual radar transects.

"(i) climate forcing, (ii) model parameterisation, (iii) bedrock and geothermal heat flux." I agree with the assessment that these three processes affect ice-sheet internal structure and like the way you have gone about targeting them individually!

Thank you.

The modelled isochrone elevations discussed above were computed on the basis of transient snapshots of local velocity and topography fields and show a good match to observed isochrone elevations." More details are needed here. What is the initial state? How much does this vary from present day? How does velocity vary in time? I assume very little? I realize these details are probably given in Sutter 2019, but it would be good to give a brief summary here. Especially given the next section of text.

We expanded the model section 2.2 considerably which now contains further information on the model initialisation as well as the tuning targets. Please see revised ms with track changes for a full account.

Use of pd or pd-pd – which one do you want to use?

Thank you for pointing this out. We now call the three cases pal, pd-pal and pd. Please note, that pd now refers to a parameter ensemble instead of to a single simulation.

"ice-sheet model parameterisations" I don't feel this point is sufficient explored in the text. What parameterisations are you referring to? Have you used models with difference ice rheology parameters, etc?

We totally agree that this was under-explored in the manuscript. We expanded the isochrone analysis by a full present day parameter ensemble, which is illustrated in figure 7 and figure 11. Please also note the revised structure of section 3 which now consists of subsections

3.1: Dome C - evaluating the paleoclimate forcing (i)

3.2: Dome C - impact of paleo spin-up and model parameterisation on simulated isochrone elevation (ii)

3.3: Impact of lower boundary conditions on isochrone simulated isochrone elevation (iii)

3.4: Caveats to modelling isochrones with large scale ice-sheet models.

Section 3.2 provides a detailed discussion of the parameter ensemble.

"All isochrone elevations simulated in the pd case are unrealistic but also show a substantial spread for the parameter range tested here (see Section 2.2). Increasing either the basal friction (via the till friction angle) or the parameter controlling the sliding (via q) shifts the isochrone elevation by almost a third of the local ice thickness. However, even for parameter sets which lead to a growing ice sheet under present-day climate conditions (corresponding with an ice sheet model parameterization which leads to high basal drag and slower vertical and horizontal ice advection) the simulated position of the 96 ka isochrone is well below the observed elevation. This shows, that it is only possible to simulate realistic isochrone elevations, while achieving an overall ice sheet shape in agreement with present-day observations, if one takes into account the paleo-evolution of the ice sheet."

Figures 7 and 8 – these figures are under utilized and they are very few references to them in the text. I suggest you pick out some more details that the reader may find interesting and include them in the main text.

We agree. Figures 7 and 8 (now figure 8 and 9) are now revised and referenced more often. Figure 8 now also includes an illustration of the effect of different geothermal heat flux choices on isochrone elevation and basal melt.

"Basal melting at the bed of the ice along the radar tracks is unfortunately unknown" Is there any evidence of water in radar profiles, i.e. flat bright reflectors, or isochrones that are drawn down and intersect the bed, that may suggest melting?

Yes, e.g. Passalacqua et al. (TC 2017) use the higher reflectivity of wet bedrock to reconstruct geothermal flux around Dome C. Fujita et al., 2012 (doi:10.5194/tc-6-1203-2012) provide an analysis of thermal vs. frozen bed conditions along a transect between Dome Fuji and EDML which could be used in the future to locally attribute mismatches to inconsistencies in basal melting between model and observations. Isochrones that are drawn down and intersect with the bed might be another possible indicator. However, the available dated isochrones we have access to are not deep enough for this to occur. It would be interesting, as you suggest, to analyze the deepest isochrones with respect to such dips and try to find a correlation to larger misfits in our modelled data. We will keep this in mind for the future. In fact, it has already been discussed as one potential metric derived from the envisaged compilation of AntArchitecture. We amended the text accordingly.

**"In the future observation-based estimates of the presence or absence of basal melt (e.g. Fujita et al., 2012; Passalacqua et al., 2017; Karlsson et al., 2018) could be utilised to locally attribute mismatches between modelled and observed isochrone elevations to inconsistencies in basal melting between model and observations."**

synoptic activity can dominate the spatial and temporal variability in precipitation" I don't know too much about spatial and temporal patterns in accumulation, but is there significant inter-annual variability in regional climate models in these regions that could be used to give some additional bounds on the changes that could be expected?

We do not know about the inter-annual variability in regional climate models but it has been shown that in DML high-precipitation events can lead to biases in ice cores (Schlosser et al., 2010) and while these events are rare can still dominate the accumulation regime (Reijmer and van den Broeke, 2003). One decisive question for paleo ice sheet modelling would be, whether these high precip. events work similar in glacials and interglacials (which would maybe allow for a simple temperature-precip. relationship as we use it in the forcing of our ice sheet model simulations) or whether there is a qualitative shift in the occurrences of synoptic scale systems e.g. in the Weddell Sea (which would make the temperature-precip. scaling approach more unrealistic). It is not possible to make a statement with respect to regional accumulation regime based on our isochrone modelling, beyond the fact that a strong temperature-precip. relationship works well around EDML (ca. 8%/K compared to 5-6% at Dome C/Dome Fuji).

identify past accumulation patterns" How reliant is this on having additional data from ice-cores or good climate models?

We reworded this (please also note the new structure of the Discussion section). It is absolutely true, that our paleo climate forcing depends on the quality of both the climate model input as well as the degree of correlation between large scale climate variations and the scaling we perform based on the Dome C deuterium data (see previous response with respect to DML). We removed the sentence stating: "We identify past accumulation patterns". The respective paragraph now reads:

"We are able to reconstruct most large-scale englacial layer features of the observed isochrones and show that it is possible to simulate the observed internal structure of the Antarctic Ice Sheet even at coarse resolution. We identify mismatches between modelled and observed isochrone elevations. This can be traced back to the transient paleoclimate forcing employed in our model runs which makes use of a linear paleo-temperature-precipitation relationship. The forcing is constructed by ice core reconstructions in combination with paleoclimate model data. This does not take into account the spatial heterogeneity of paleo temperature-precipitation relationships and effects of synoptic variability. Our isochrone modelling efforts therefore motivate the use of a regionally refined

temperature-precipitation scaling to improve paleo ice-sheet simulations and consequently the paleo spinup for model based future projections."

While analysing the match of an ISM simulation with the internal stratigraphy is not as straight forward as using surface observables, it could improve both paleo ice-sheet reconstructions as well as sea level projections due to more realistic initial ice-sheet configurations." It would be good to detail what impacts (if any) an incorrect internal ice-sheet stratigraphy would have on future projections.

When the tuning of an ice sheet model is restricted to present day observations there are several factors which can lead to a biased future model behaviour. The parameters relevant e.g. for basal sliding will be set based on uncertainties in geothermal heat flux, present day climate forcing (ocean+surface) and the ice sheet's temperature field. Even if using inversion techniques this will lead to a prescribed distribution of basal drag which might create the correct present day surface elevation but is based on uncertain input fields. This will then lead to a bias in the future flow patterns of the ice sheet. Quantifying this effect with the help of isochrone-matching will be an interesting task but is beyond the scope of this study. We discuss this explicitly in the introduction of the manuscript:

"Using the above-mentioned tuning targets allows for simulation of an ice-sheet in line with the present-day observed surface properties and proxy data from the past millennia. However, the notion that a good fit to spatial datasets of the Common Era and proxy targets in the past guarantees the model's ability to respond accurately to future climate changes is debatable. Due to the complexity of ice-sheet-climate interactions, lack of spatial data sets for past climate states and uncertainties in paleo-proxy based ice-sheet and sea level reconstructions, it is still challenging to create a proper initial ice-sheet configuration from which its future evolution can be adequately simulated (Seroussi et al., 2019, 2020). For example, tuning the ice-sheet to the observed present state (e.g. via inversion for basal drag) by matching the current ice-sheet topography and surface flow does not guarantee the accurate reproduction of e.g. internal flow, ice temperature distribution and basal friction. It also could lead to overfitting of parameters relevant to ice flow within the scope of uncertain boundary conditions such as geothermal heat flux, sub-shelf ocean temperatures, and surface mass balance. Without inversion the modelled present-day topography can differ from the observed state by several hundred meters of ice thickness on which basis it is difficult to interpret model-based sea level projections. Fundamentally, every ISM application is an ill-posed problem with non-unique solutions. Therefore, overfitting to a set of observables could lead to an initial ice-sheet configuration dominating the projected response to the applied climate forcing (Seroussi et al., 2019), especially over decadal to centennial timescales. Discrepancies in the initial state with respect to the actual real world ice-sheet can propagate and multiply during the model simulation due to the intrinsic nonlinearities of the system. Even a near-perfect match to present-day 2D or 1D observable state variables can conceal overfitting of the model due to weakly constrained boundary conditions, e.g. uncertainties in the climate forcing or geothermal heat flux (Burton-Johnson et al., 2020; Talalay et al., 2020). To counter the effects of overfitting, promising attempts to improve the initialisation of ice sheet models have been made which involve transient inversions of multiple surface elevation observations over time (Goldberg et al., 2015) albeit only on the regional scale and limited to the extent of the sattelite record."

Figures: Most figures and captions need some attention to improve their readability. At present they have the potential to be really good, but need a little more work. I have included comments in the attached PDF.

Thank you for your positive assessment. We modified all figures according to your and the other Reviewers suggestions. We hope that we were able to improve the readability and the appeal of the figures thanks to your suggestions.

One reoccurring issue is the use of compound adjectives: ice-sheet model, present-day accumulation, etc. This should be addressed consistently throughout the text.

Done.

[revised manuscript text omitted]

---

## Referee Report (RR1)

[referee-annotated manuscript omitted]

---

## Referee Report (RR2)

Review of "*Investigating the internal structure of the Antarctic Ice Sheet: the utility of isochrones for spatio-temporal ice sheet model calibration*" by Sutter, Fischer, and Eisen (2021)

**From previous review**: *The manuscript by Sutter et al. (2021) shows how the ice sheet internal layer structure can be exploited to understand and diagnose ice sheet model output. The authors present a clear case for the utility of comparing isochrones derived from observations and ice sheet model simulations to determine ice sheet model performance, particularly highlighting where we need:*

*(1) Better constraints on boundary conditions (e.g. bed topography; geothermal heat flux);*
*(2) Better constraints on climate forcings (e.g. spatial variation in paleo accumulation rates);*
*(3) Better constraints on ice sheet model parameters (e.g. basal drag over marine sectors of the ice sheet);*
*(4) Long term simulations to adequately represent 3D flow fields and ice sheet geometries.*

*The diagnostic method presented in this manuscript (i.e. use of particle tracer method) is freely available and can be readily applied to any ice sheet model output, making this diagnostic tool accessible for ice sheet modellers.*

*The manuscript addresses a highly relevant scientific question, especially with the work of the AntArchitecture project. To the best of my knowledge the concept is novel, and the scope of the model simulations and comparison with observations is appropriate to support the interpretations and conclusions, and to demonstrate broad applicability of the method to the ice sheet modelling community. Overall, this is a worthwhile study that is certainly within the scope of TC.*
* * *
The authors have addressed my concerns from the previous review. The structure of sections 3-5 has been greatly improved. The introduction of the RMSD analysis to benchmark the simulated isochrone elevations against the observed tightens the results/analysis nicely.

I have minor comments below that should be addressed before publication.

**Minor comments**

P1L9 (and throughout manuscript). Remove the comma before "that"

P1L9-11. Depending on word limits, you may want to consider adding to the abstract the fact that calibrating to present-day yields isochrone elevations that are often substantially more inaccurate, as per your point on L328-330.

P1L21. Add Edwards et al. (2021) to the citations

Edwards, T. L., Nowicki, S., Marzeion, B., Hock, R., Goelzer, H., Seroussi, H., ... & Zwinger, T. (2021). Projected land ice contributions to twenty-first-century sea level rise. Nature, 593(7857), 74-82.

P3L81. "...as detailed as possible" >> "...in as detailed a way as possible"

P3L85. "shortly" >> "briefly" (or delete)

P4L94 (and elsewhere in manuscript). "Ice-sheet model" >> "ISM"?

P7L161. "where" >> "were"

P9L213. "accurate enough" >> "within expected uncertainty tolerances"? I.e. from observations?

P9L222-224. "In regions where..." It'd be good to be more explicit in these sentences about which distances along the transect in figure 3 you're referring to.

Also, is the divergence between 200 and 400 km of the 5 ka 10 k line in panel B related to spatial gradients in ice-flow and topography, as you suggest? If so, it's hard to see why this happens here, but not later in the transect where there are significant gradients in topography (e.g. between 400 and 500 km).

Figure 3. Should "5 ka 10 k" in the figure legend be "5 ka 5 k"? The caption says that seeding was carried out with 200 000 and 5 000 tracers.

Figure 4 caption. "blow up" >> "magnification"

P12L276. Remove wayward parenthesis ")"

Figure 5. Topography color scale could be moved to left panel.

P13L280. "%3K$^{-1}$" >> "3 %K$^{-1}$"

P13L295. Check citation style with van Wessem reference

Figure 6. I didn't understand where the 7 %K$^{-1}$ scaling came from? Perhaps add a reason why in the caption, or add it to the list of experiments in section 2.2

P15L322. "relatively small". Indicate % difference

P15L331-340. This is a really important point and it'd be good to see a follow-up on this point in the conclusion. E.g. how do we practically address the problems you've highlighted? Should long-term ice sheet model simulations all be starting from a paleo spin-up? Obviously this is

unfeasible in many cases, but it would be good to move towards a standard approach/methodology, or at least have a way forward to address some of these issues.

Figure 7 caption. Add % to the RMSD values in parentheses. Is it "root mean square error" or "root mean square difference"?

Figure 8. Can you modify the y-axis limits in panel B) so we can see all of the red line?

P18L389. "diversions" >> "divergence"

P18L400-401. "This could be due to the heuristics involved in determining the yield stress in subglacial basins." This deserves elaboration here or in the conclusions.

P19L412-414. Given your conclusions about the issues with basal friction, can you recommend a more appropriate friction law to use?

P20L420. "underlines" >> "agrees with"?

P20L443. "...using the state of the art today" >> "...using the state of the art models today"

Figure 10. Add y-axis labels to the panels in C)

P24L484. "paloe" >> "paleo"

P24L494-500. Mention that improved GHF estimates, as per Stål et al. (2020) may reduce uncertainty.

Stål, T., Reading, A. M., Halpin, J. A., & Whittaker, J. M. Antarctic geothermal heat flow model: Aq1. Geochemistry, Geophysics, Geosystems, e2020GC009428.

---

## Editor Decision (ED1)

I would like to thank the two reviewers for providing feedback on the resubmitted version of the article by Sutter et al. titled 'Investigating the internal structure of the Antarctic Ice Sheet: the utility of isochrones for spatiotemporal ice sheet model calibration'.

Both reviewers are satisfied that the authors have addressed the major comments from their initial reviews, but both have also submitted some minor comments that should be addressed prior to publication. In addition to the comments from the reviewers, I list below a number of points to consider, which will improve the clarity of the article for the reader. I recommend that the article should be published once these minor issues are addressed.

Kind regards,

Pippa Whitehouse (Editor)

Points to be addressed:

1) As you make your final revisions, I encourage you to look for places where the text can be made more concise, or sentences shortened. Long sentences require careful punctuation to ensure they are correctly interpreted. Short sentences are much clearer!

2) Ensure that acronyms are defined at their first usage, and that they are used consistently throughout the remainder of the text

3) Check the format of in-text citations

4) When using the term 'topography', clarify whether you are referring to the surface or the bed of the ice sheet

5) It is a little unclear whether some of your results are derived from the 2 Ma experiment, or whether this experiment is simply used to initiate the 220 ka experiments and all results shown are derived from the 220 ka experiments. It would be useful to clarify this in section 2.2

6) Your methodology provides an estimate of the normalised elevation of each isochrone above the bed. However, radar systems provide an estimate of the depth of an isochrone below the ice surface. Any uncertainty on total ice thickness/bed elevation will impact on your ability to compare modelled and observed isochrone positions. Please briefly comment on this issue.

7) I do not have a strong opinion on the length or content of your conclusions (comment from reviewer 1) but the logic of your argument could be clearer in a few places (e.g. lines 491-, 495-)

8) Figures: please check the following points in relation to all figures:
   a) Colour scales are included where relevant
   b) The caption describes all features shown in the figures
   c) Somewhere, state the projection used to define the northing/easting values
   d) Ensure that all place names mentioned in the text are indicated on a figure
   e) Check the location of all transect plots is clear (e.g. this is not the case for figure 3)
   f) Define all lines in the legend/caption, including the lines representing the ice sheet surface
   g) Ensure that all sub-plots are labelled
   h) Ensure that all axes are labelled and that labels are legible
   i) Use the first sentence of the caption to identify what is distinct about each figure, e.g. for figure 8, 'The influence of geothermal heat flux on predictions of isochrone elevation'
   j) Quantify information shown in the figures when describing or discussing results in the text

---

## Author Response (AR2)

We are grateful to the reviewer for taking the time to re-review our manuscript and the positive final assessment. We refer to the marked up manuscript for the changes based on the review.

We are grateful to the reviewer for taking the time to re-review our manuscript and the positive final assessment. In the following we provide a point-by-point response (reviewer comments in black, replies in blue, changes to the text in non-serif) and refer to the attached track-changes document for the changes made to the manuscript (new text in blue, modified text in red).

Review of "Investigating the internal structure of the Antarctic Ice Sheet: the utility of isochrones for spatio-temporal ice sheet model calibration" by Sutter, Fischer, and Eisen (2021)

From previous review: The manuscript by Sutter et al. (2021) shows how the ice sheet internal layer structure can be exploited to understand and diagnose ice sheet model output. The authors present a clear case for the utility of comparing isochrones derived from observations and ice sheet model simulations to determine ice sheet model performance, particularly highlighting where we need:

(1) Better constraints on boundary conditions (e.g. bed topography; geothermal heat flux);
(2) Better constraints on climate forcings (e.g. spatial variation in paleo accumulation rates);
(3) Better constraints on ice sheet model parameters (e.g. basal drag over marine sectors of the ice sheet);
(4) Long term simulations to adequately represent 3D flow fields and ice sheet geometries.
The diagnostic method presented in this manuscript (i.e. use of particle tracer method) is freely available and can be readily applied to any ice sheet model output, making this diagnostic tool accessible for ice sheet modellers.

The manuscript addresses a highly relevant scientific question, especially with the work of the AntArchitecture project. To the best of my knowledge the concept is novel, and the scope of the model simulations and comparison with observations is appropriate to support the interpretations and conclusions, and to demonstrate broad applicability of the method to the ice sheet modelling community. Overall, this is a worthwhile study that is certainly within the scope of TC.

The authors have addressed my concerns from the previous review. The structure of sections 3-5 has been greatly improved. The introduction of the RMSD analysis to benchmark the simulated isochrone elevations against the observed tightens the results/analysis nicely.

Again, we thank the reviewer for the positive final assessment!

I have minor comments below that should be addressed before publication.
Minor comments

done

P1L9 (and throughout manuscript). Remove the comma before "that"

done

P1L9-11. Depending on word limits, you may want to consider adding to the abstract the fact that calibrating to present-day yields isochrone elevations that are often substantially more inaccurate, as per your point on L328-330.

We thank the reviewer for this suggestion. We tried this, but we already maxed out the abstract word count for TC-manuscripts unfortunately.

P1L21. Add Edwards et al. (2021) to the citations
Edwards, T. L., Nowicki, S., Marzeion, B., Hock, R., Goelzer, H., Seroussi, H., ... & Zwinger, T. (2021). Projected land ice contributions to twenty-first-century sea level rise. Nature, 593(7857), 74-82.

Done. We were not sure where best to place this reference as Edwards et al. use statistical emulators of existing models. We added a general reference on page 2 l24.

P3L81. "...as detailed as possible" >> "...in as detailed a way as possible"

done

P3L85. "shortly" >> "briefly" (or delete)

deleted

P4L94 (and elsewhere in manuscript). "Ice-sheet model" >> "ISM"?

done

P7L161. "where" >> "were"

Thanks for spotting this! Done.

P9L213. "accurate enough" >> "within expected uncertainty tolerances"? I.e. from observations?

Done. We modified the sentence "[…] accurate enough (small deviations compared to the model uncertainties with respect to observations) for the mostly relatively slow ice flow in East Antarctica.

P9L222-224. "In regions where..." It'd be good to be more explicit in these sentences about which distances along the transect in figure 3 you're referring to.
Also, is the divergence between 200 and 400 km of the 5 ka 10 k line in panel B related to spatial gradients in ice-flow and topography, as you suggest? If so, it's hard to see why this happens here, but not later in the transect where there are significant gradients in topography (e.g. between 400 and 500 km).

Thanks for spotting this. We amended the manuscript by the sentence This is shown along kms 400-200 in Figure 3 B where elevated surface velocities closer to the coast lead to more pronounced deviations for a coarse seeding strategy.

Figure 3. Should "5 ka 10 k" in the figure legend be "5 ka 5 k"? The caption says that seeding was carried out with 200 000 and 5 000 tracers.

Yes. Thanks for spotting this!

Figure 4 caption. "blow up" >> "magnification"

done

P12L276. Remove wayward parenthesis ")"

Figure 5. Topography color scale could be moved to left panel.

We decided to leave the color scale in panel B as panel A would otherwise be too crowded.

P13L280. "%3K$_{-1}$" >> "3 %K$_{-1}$"

Thanks for spotting this, done.

P13L295. Check citation style with van Wessem reference

done

Figure 6. I didn't understand where the 7 %K$_{-1}$ scaling came from? Perhaps add a reason why in the caption, or add it to the list of experiments in section 2.2

Yes this is confusing. We use 7% here as it best matches the reconstruction of Cavitte et al 2018 after bias correction. We did not carry out a paleo simulation with 7% however.

P15L322. "relatively small". Indicate % difference

done

P15L331-340. This is a really important point and it'd be good to see a follow-up on this point in the conclusion. E.g. how do we practically address the problems you've highlighted? Should long-term ice sheet model simulations all be starting from a paleo spin-up? Obviously this is unfeasible in many cases, but it would be good to move towards a standard approach/methodology, or at least have a way forward to address some of these issues.

We thank the reviewer for this comment, as we also think that this touches on a key issue in current modelling efforts: a more structured approach to ice sheet spin-ups. As the Reviewer stated, every research hypothesis and model setup will lead to a specialised model spin-up. However, for a general "species" of model studies such as paleo-simulations or long term projections common spin-up standards should be formulated. A more detailed assessment of the status quo and the way forward is a little beyond the scope of this manuscript which is already lengthy. But we will keep this in mind for the planned model intercomparison project. Thanks again for pointing this out!

Figure 7 caption. Add % to the RMSD values in parentheses. Is it "root mean square error" or "root mean square difference"?

Done and corrected, we mean RMSD

Figure 8. Can you modify the y-axis limits in panel B) so we can see all of the red line?

Done. We thank the reviewer for commenting on this, as during re-plotting of the data we realised that we used an experiment with a coarse seeding strategy to plot the data for this figure

P18L389. "diversions" >> "divergence"

done

P18L400-401. "This could be due to the heuristics involved in determining the yield stress in subglacial basins." This deserves elaboration here or in the conclusions.

done

P19L412-414. Given your conclusions about the issues with basal friction, can you recommend a more appropriate friction law to use?

This is something we want to investigate in a planned research project ☺

P20L420. "underlines" >> "agrees with"?

done

P20L443. "...using the state of the art today" >> "...using the state of the art models today"

done

Figure 10. Add y-axis labels to the panels in C)

done

P24L484. "paloe" >> "paleo"

Thanks for spotting this! Done.

P24L494-500. Mention that improved GHF estimates, as per St l et al. (2020) may reduce uncertainty.
St l, T., Reading, A. M., Halpin, J. A., & Whittaker, J. M. Antarctic geothermal heat flow model: Aq1. Geochemistry, Geophysics, Geosystems, e2020GC009428.

Done.

---

## Editor Decision (ED2)

**Comments on "Investigating the internal structure of the Antarctic Ice Sheet: the utility of isochrones for spatiotemporal ice-sheet model calibration" by Sutter et al.**

Thank you for submitting a revised version of your article. A number of the comments raised by one of the reviewers and the editor have not been fully addressed. These are listed below. Line numbers refer to the most recent non-track-changed version of the article. In cases where an example is given, please check for other instances throughout the manuscript. All points are minor but should be addressed prior to publication.

Pippa Whitehouse (Editor)

**RC = reviewer comment**
**EC = editor comment**
non-bold – editor's comment explaining why the points has not been sufficiently addressed

**RC: P1L9 (and throughout manuscript). Remove the comma before "that"**
Commas have not been removed on lines 9, 259, 273, 288, 302, and 322

**RC: P9L222-224. … it's hard to see why this happens here, but not later in the transect where there are significant gradients in topography [comment relates to figure 3]**
Please address the reviewer's query about why there is no divergence in modelled isochrones in the region of steep topography between 400 and 500 km on the transect

**RC: P12L276. Remove wayward parenthesis ")"**

**RC: Figure 5. Topography color scale could be moved to left panel.**
Please do not locate the colour scale for one plot within a different plot, this is confusing

**RC: Figure 6. I didn't understand where the 7 %K$^{-1}$ scaling came from? Perhaps add a reason why in the caption, or add it to the list of experiments in section 2.2**
You note that this point is confusing and provide an explanation in your response to the reviewer. Please also include relevant information in the revised article to explain this point to the reader

**RC: comment on spin-up approach:** you mention this article is not the place for a detailed discussion of the best approach to ice-sheet model spin-up. However, it would be useful if you could mention in the conclusions why this point is important, i.e. re-iterate your point on line 335 that ice sheet initial state can significantly affect its future behaviour over centennial and decadal timescales.

**RC: P19L412-414. Given your conclusions about the issues with basal friction, can you recommend a more appropriate friction law to use?**
You mention that this is a proposed area of future research. It would be useful for the reader if you could briefly outline the alternative approaches that could be adopted

**RC: Figure 10. Add y-axis labels to the panels in C)**

**EC: Ensure that acronyms are defined at their first usage, and that they are used consistently throughout the remainder of the text**
For example, 'AIS' is not defined on line 6

**EC: Check the format of in-text citations**
For example, see line 160

**EC: When using the term 'topography', clarify whether you are referring to the surface or the bed of the ice sheet**
For example, see caption to figure 1

**EC: It is a little unclear whether some of your results are derived from the 2 Ma experiment, or whether this experiment is simply used to initiate the 220 ka experiments and all results shown are derived from the 220 ka experiments. It would be useful to clarify this in section 2.2**
No edits were made to clarify this point. In particular, it is unclear whether the 'pal' results are derived from models run over a mixture of 2 Ma and 220 ka, and whether the 'pd-pal' results are based on present-day snapshots of models run over a mixture of 2 Ma and 220 ka

**EC: Your methodology provides an estimate of the normalised elevation of each isochrone above the bed. However, radar systems provide an estimate of the depth of an isochrone below the ice surface. Any uncertainty on total ice thickness/bed elevation will impact on your ability to compare modelled and observed isochrone positions. Please briefly comment on this issue.**
No response to this query, consider whether it warrants a comment within the manuscript

**EC: logic of your argument could be clearer in a few places (e.g. lines 491-, 495-)**
The logic behind the final sentence of the third bullet point in the conclusions is unclear. There is a jump in logic between the first and second sentences in the fourth bullet point in the conclusions.

**EC: Figures: please check the following points in relation to all figures:**
a) **Colour scales are included where relevant**
   for example, figure 1, figure 2, figure 9
b) **The caption describes all features shown in the figures**
   please check the accuracy of all captions. In some cases, captions do not agree with information in the figure, e.g. figure 8 refers to Purucker (2013) data which is not shown in the figure, it also refers to 'thin dashed lines' in panel B, which are not visible.
c) **Somewhere, state the projection used to define the northing/easting values**
d) **Ensure that all place names mentioned in the text are indicated on a figure**
   for example, the reader is referred to figure 1 to locate Dronning Maud Land and George V coast (line 108) but these locations are not labelled on figure 1
e) **Check the location of all transect plots is clear (e.g. this is not the case for figure 3)**
   in particular, it is not always clear which end of the transect is defined as 0 km
f) **Define all lines in the legend/caption, including the lines representing the ice sheet surface**
   in several plots multiple lines are plotted at ~3000 m elevation. I think these represent the surface of the ice sheet, but it is not clear what the difference is between the lines
g) **Ensure that all sub-plots are labelled**
   for example, figure 7
h) **Ensure that all axes are labelled and that labels are legible**
   for example, figure 7 (x-axis, right-hand plot), figure 10C (both axes)

Additional editor comment: it is unclear what some of the numbers refer to in the edits to line 323, e.g. "pal 4.8 and pd 20"

---

## Author Response (AR3)

Comments on "Investigating the internal structure of the Antarctic Ice Sheet: the utility of isochrones for spatiotemporal ice-sheet model calibration" by Sutter et al.
Thank you for submitting a revised version of your article. A number of the comments raised by one of the reviewers and the editor have not been fully addressed. These are listed below. Line numbers refer to the most recent non-track-changed version of the article. In cases where an example is given, please check for other instances throughout the manuscript. All points are minor but should be addressed prior to publication.
Pippa Whitehouse (Editor)

We sincerely apologize for omitting some of the reviewer comments and the changes to the manuscript you suggested. We hope we can rectify this embarrassing omission with the newly revised version of the manuscript. Please find point-by-point replies to the remaining comments in Blue and changed text in Red.

RC = reviewer comment
EC = editor comment
non-bold – editor's comment explaining why the points has not been sufficiently addressed

RC: P1L9 (and throughout manuscript). Remove the comma before "that"
Commas have not been removed on lines 9, 259, 273, 288, 302, and 322

Done

RC: P9L222-224. … it's hard to see why this happens here, but not later in the transect where there are significant gradients in topography [comment relates to figure 3]
Please address the reviewer's query about why there is no divergence in modelled isochrones in the region of steep topography between 400 and 500 km on the transect

We rephrased the sentence (now p9l225-226) so that it only refers to gradients in ice flow and not topography as tracers are advected via the 3D velocity field, thus deviations in isochrone elevations for different tracer seeding parameters are due to gradients in the velocity field. The modeled isochrone elevation is affected by the bedrock elevation and gradients in the modelled bedrock contour can deviate from the real bedrock landscape considerably. The region between transect km's 400 and 500 is a good example where high-res radar data shows large local variations in bedrock elevation while the coarse resolution model bedrock elevation is essentially flat (and therefore does not affect the isochrone elevation much for different tracer seeding strategies). A more detailed analysis would be necessary to identify at which point velocity gradients adversely affect computed isochrone elevations for coarse seeding strategies. We will keep this in mind for the future.

RC: P12L276. Remove wayward parenthesis ")"

Done.

RC: Figure 5. Topography color scale could be moved to left panel.
Please do not locate the colour scale for one plot within a different plot, this is confusing

We moved the colour-scale to the corresponding plot.

RC: Figure 6. I didn't understand where the 7 %K-1 scaling came from? Perhaps add a reason why in the caption, or add it to the list of experiments in section 2.2

You note that this point is confusing and provide an explanation in your response to the reviewer. Please also include relevant information in the revised article to explain this point to the reader

We apologize for this omission. We decided to change this figure so it shows the forcing for 6 %K-1 scaling which is consistent with the experiments and hopefully removes the source of confusion. We modified the figure caption accordingly.

RC: comment on spin-up approach: you mention this article is not the place for a detailed discussion of the best approach to ice-sheet model spin-up. However, it would be useful if you could mention in the conclusions why this point is important, i.e. re-iterate your point on line 335 that ice sheet initial state can significantly affect its future behaviour over centennial and decadal timescales.

We amended the 3ʳᵈ bullet point in the conclusions by the following paragraph:

Even small biases (e.g. due to overfitting against uncertain input fields) in the initial model state can impact ice sheet dynamics and therefore estimates of future ice sheet sea level contributions. We make the case that the paleo-evolution of an ice sheet should be considered both for reconstructions as well as projections of ice sheet changes and that isochrones are ideally suited for this purpose.

RC: P19L412-414. Given your conclusions about the issues with basal friction, can you recommend a more appropriate friction law to use?
You mention that this is a proposed area of future research. It would be useful for the reader if you could briefly outline the alternative approaches that could be adopted

We now mention potential alternative avenues to model basal friction in the conclusions, p25-l510-512:

For example, the impact of different calibrations of basal drag on modelled isochrone elevations, such as inversion methods based on surface elevation and ice flow, could be elucidated in such an intercomparison.

RC: Figure 10. Add y-axis labels to the panels in C)

Done

EC: Ensure that acronyms are defined at their first usage, and that they are used consistently throughout the remainder of the text
For example, 'AIS' is not defined on line 6

Done

EC: Check the format of in-text citations
For example, see line 160

Done

EC: When using the term 'topography', clarify whether you are referring to the surface or the bed of the ice sheet
For example, see caption to figure 1

Done

EC: It is a little unclear whether some of your results are derived from the 2 Ma experiment, or whether this experiment is simply used to initiate the 220 ka experiments and all results shown are derived from the 220 ka experiments. It would be useful to clarify this in section 2.2
No edits were made to clarify this point. In particular, it is unclear whether the 'pal' results are derived from models run over a mixture of 2 Ma and 220 ka, and whether the 'pd-pal' results are based on present-day snapshots of models run over a mixture of 2 Ma and 220 ka

We agree that this is not clear from section 2.2. All simulations were carried out for this manuscript except for the geothermal heat flux ensemble using a 3%/K temperature precipitation scaling which was simply taken from Sutter et al. 2019. We now begin the section with the following paragraph (p6l157-165).:

"In order to compute Antarctic isochrones, time-resolved 3D velocity data as well as the transient ice-sheet geometry (ice thickness and bedrock topography) are necessary. We therefore ran a paleoclimate model ensemble covering the last 220 ka. All 220 ka simulations were initialised from the 220 ka output of a 2 Ma long simulation in Sutter et al. (2019). We also make use of four simulations from Sutter et al. (2019) which are based on four different geothermal heat flux fields (Shapiro and Ritzwoller, 2004; Purucker, 2013; An et al., 2015; Martos et al., 2017). In addition to the 220 ka paleo-ensemble we carried out a present-day equilibrium ensemble to assess the impact of the missing paleo-spinup as well as different model parameterisations on the computed isochrone elevations."

And follow with

"Isochrone elevations (see 2.3) are computed on the basis of: 1) full paleo-ISM runs (called pal) in which model integration starts from the 220 ka time slice of a 2 Ma simulation (Sutter et al., 2019), 2) the present- day snapshot of the 220 ka simulation (pd-pal), 3) a present-day equilibrium ensemble (pd) with an integration time of 2000 years following a thermal spinup using a fixed ice sheet geometry for 200 ka. The 2 Ma simulations in Sutter et al. (2019) are initialised at 2 Ma BP from a present-day ice sheet geometry. Isochrone evolution starts at the isochrone's respective age (see Table 1) in the past and follows the computed transient ice flow. In the present-day snapshot (pd-pal) and present-day equilibrium ensemble (pd) isochrones evolve on the basis of the simulated present-day flow (constant velocity field). "

EC: Your methodology provides an estimate of the normalised elevation of each isochrone above the bed. However, radar systems provide an estimate of the depth of an isochrone below the ice surface. Any uncertainty on total ice thickness/bed elevation will impact on your ability to compare modelled and observed isochrone positions. Please briefly comment on this issue.

No response to this query, consider whether it warrants a comment within the manuscript

This is true and so far, we only shortly mention this matter in the conclusions (see next comment). One could potentially mitigate this issue by tracing the elevation of the individual tracers relative to the surface of the ice which generally better agrees with observations than the bed (at least for present day). In PISM the vertical coordinate is defined relative to the base of the ice and a coordinate transformation would be necessary to track the ice relative to the surface. However, we deemed it to be more consistent to trace the ice trajectories accordingly to the native geometry of the ice sheet model. We added a sentence in section 2.3 (Lagrangian tracer advection) stating:

"Here, it is important to note that observed isochrone elevations are usually defined as relative to the ice surface whereas we compute the isochrone elevation above the ice bed. Any deviations in the modelled ice bed with respect to observations will therefore imprint on the modelled isochrone elevation. Therefore, any comparison between modelled and observed isochrone elevations will be most meaningful along transects with small deviations between the observed and modelled bedrock elevation."

EC: logic of your argument could be clearer in a few places (e.g. lines 491-, 495-)
The logic behind the final sentence of the third bullet point in the conclusions is unclear.

We agree and reformulate the phrase as follows:

"Varying the parameter space relevant for basal sliding for a set of parameters that produce an equilibrium sea-level equivalent ice volume of the Antarctic Ice Sheet within ±2 m of present-day observations cannot remedy this mismatch"

There is a jump in logic between the first and second sentences in the fourth bullet point in the conclusions.

We agree and reformulate the paragraph as follows:

"When using isochrones as a tuning target for paleo-ISMs, two key uncertainties prove difficult to account for: 1) geothermal heat flux fields remain poorly constrained (new, e.g. Stal et al. 2021, and upcoming datasets might reduce this uncertainty) and can have a strong influence on isochrone elevations. 2) Uncertain bedrock elevation in regions with gaps in radar surveys affect modelled isochrone elevations especially for isochrones close to bedrock. However, for areas covered by high resolution radar transects, this aspect can be quantified by comparison to the model bedrock elevation. Combining isochrone elevations, present-day observables and paleo proxy data in the calibration of ice sheet model setups helps to mitigate aforementioned uncertainties and prevent overfitting."

EC: Figures: please check the following points in relation to all figures:

a)      Colour scales are included where relevant
for example, figure 1, figure 2, figure 9

Done

b)      The caption describes all features shown in the figures

please check the accuracy of all captions. In some cases, captions do not agree with information in the figure, e.g. figure 8 refers to Purucker (2013) data which is not shown in the figure, it also refers to 'thin dashed lines' in panel B, which are not visible.

Done. The thin dashed lines described in the caption for figure 8 B) where present in an older version of the figure which we decided not to use. We removed the sentence referring to the thin dashed lines.

c)      Somewhere, state the projection used to define the northing/easting values

Done in the caption of figure 1.

d)      Ensure that all place names mentioned in the text are indicated on a figure
for example, the reader is referred to figure 1 to locate Dronning Maud Land and George V coast (line 108) but these locations are not labelled on figure 1

Done.

e)      Check the location of all transect plots is clear (e.g. this is not the case for figure 3)
in particular, it is not always clear which end of the transect is defined as 0 km

In the caption of figure 3 we note that transect CEA-10 is depicted and refer to figure 1,10,11 for the location. We agree that the location is not given in Figure 11 and a little hard to identify in Figure 1. Thus we now only point the reader to Figure 10 :

"[…] see Figure 10 for the location of transect CEA-10 "

 The direction of the transect in Figure 3 is identical to the one in Figure 10 and 11.

f)      Define all lines in the legend/caption, including the lines representing the ice sheet surface
in several plots multiple lines are plotted at ~3000 m elevation. I think these represent the surface of the ice sheet, but it is not clear what the difference is between the lines

Yes. Different surface elevations refer to different model runs (i.e. for different GHF input fields). Additionally, the observed surface elevation (Bedmap2) is plotted. This is now explicitly mentioned in all relevant figures.

g)      Ensure that all sub-plots are labelled
for example, figure 7

Done.

h)      Ensure that all axes are labelled and that labels are legible

for example, figure 7 (x-axis, right-hand plot), figure 10C (both axes)

Done.

Additional editor comment: it is unclear what some of the numbers refer to in the edits to line 323, e.g. "pal 4.8 and pd 20"

Thank you for spotting this. The sentence now reads:

However, it is relatively small (4.8% vs. 11.4% RMSD) in comparison to the difference between pal (4.8 % RMSD) and pd (20−56% RMSD).

---

## Author Response (AR4)

Dear Editor,

Thank you very much for your investment into the improvement of this manuscript, this is highly appreciated! We are sorry for the numerous editing mistakes from our side. We provide a point-by-point response below with the changes made according to your comments.

Thanks again for your time and work and for your positive assessment of this manuscript. We are excited to see this paper published in your Journal.

Dear authors,

Thank you for your clear responses to the issues raised in relation to the previous version of the article and for promptly submitting a revised version. A number of minor points still require clarification, primarily in relation to the figures, these are detailed below. The article may be considered to be accepted for publication once these technical issues are addressed. This is a highly novel piece of work that makes an important contribution to the validation of ice sheet model output. Thank you for submitting to The Cryosphere!

Kind regards,
Pippa Whitehouse (Editor)
* * *
Technical Issues:

Figure 1, top left plot: I think that ice surface elevation is plotted here, please clarify in the caption and consider including a colour scale

Yes. We included a colour scale and amended the caption correspondingly : "Antarctic surface (top left) and bedrock elevation from […]"

Figure 4: please explain what the black circles are on plots B) and C) and include some indication of scale on all plots, e.g. by adding units to the axes

We are sorry if this caused confusion. The caption previously read : " […] dotted areas where basal melt is simulated […]".

We now further highlight this by stating:

" […] dotted areas (filled black circles) where basal melt is simulated […]".

Figure 6: please include units (of time) on the x-axis of A) and correct the values on the x-axis of B)

Thanks, and done.

Figure 8: the caption states that this figure relates to transect DC-X57; is this the same as transect X57a mentioned in the caption to Figure 7? Check that consistent names are used for all transects

Thanks for spotting this! We changed all instances of X57a or DC-X57 to the correct full name DC-X57a (same with other transects).

Figure 9: I could not find any information, e.g. on a map, to indicate which end of the DC-Y77 transect is regarded as 0 km and which end is 100 km. The x-axis for the upper plot is not labelled: I suspect that both transects are 100 km long, but this should be stated, or both sets of x-axes labelled. In the caption, the black/grey lines at the bed are defined twice.

Yes, this is unclear. We added a sentence stating that:

"[…] for transect DC-X57a (A) and DC-Y77 (B) (orientation from left to right as depicted in Figure 7 A) […]"

Line 364: isotope -> isochrone

Done.